# Semi-supervised Dense Keypoints using Unlabeled Multiview Images

**Zhixuan Yu**
University of Minnesota
yu000064@umn.edu

**Haozheng Yu**
University of Minnesota
yu000424@umn.edu

**Long Sha**
TuSimple
long.sha@tusimple.ai

**Sujoy Ganguly**
Unity
sujoy.ganguly@unity3d.com

**Hyun Soo Park**
University of Minnesota
hspark@umn.edu

## Abstract

This paper presents a new end-to-end semi-supervised framework to learn a dense keypoint detector using unlabeled multiview images. A key challenge lies in finding the exact correspondences between the dense keypoints in multiple views since the inverse of the keypoint mapping can be neither analytically derived nor differentiated. This limits applying existing multiview supervision approaches used to learn sparse keypoints that rely on the exact correspondences. To address this challenge, we derive a new probabilistic epipolar constraint that encodes the two desired properties. (1) Soft correspondence: we define a matchability, which measures a likelihood of a point matching to the other image's corresponding point, thus relaxing the requirement of the exact correspondences. (2) Geometric consistency: every point in the continuous correspondence fields must satisfy the multiview consistency collectively. We formulate a probabilistic epipolar constraint using a weighted average of epipolar errors through the matchability thereby generalizing the point-to-point geometric error to the field-to-field geometric error. This generalization facilitates learning a geometrically coherent dense keypoint detection model by utilizing a large number of unlabeled multiview images. Additionally, to prevent degenerative cases, we employ a distillation-based regularization by using a pretrained model. Finally, we design a new neural network architecture, made of twin networks, that effectively minimizes the probabilistic epipolar errors of all possible correspondences between two view images by building affinity matrices. Our method shows superior performance compared to existing methods, including non-differentiable bootstrapping in terms of keypoint accuracy, multiview consistency, and 3D reconstruction accuracy.

## 1 Introduction

The spatial arrangement of keypoints of dynamic organisms characterizes their complex pose, providing a computational representation of the way they behave. Recently, computer vision models offer fine grained behavioral modeling through dense keypoints that establish an injective mapping from the image coordinates to the continuous body surface of humans [8] and chimpanzees [36]. These models predict the continuous keypoint field from an image, supervised by a set of densely annotated keypoints, which shows remarkable performance on real-world imagery and brings out a number of applications including 3D mesh reconstruction [54, 53, 35, 50, 7, 21], texture/style transfer [26, 37], and geometry learning [14, 1]. Nonetheless, attaining such densely annotated data is labor intensive, and more importantly, the quality of the annotations is fundamentally bounded by the visual ambiguity of keypoints, e.g., points on textureless shirt. This visual ambiguity leads to a suboptimal model when applying it to out-of-sample distributions. In this paper, we present a

35th Conference on Neural Information Processing Systems (NeurIPS 2021).

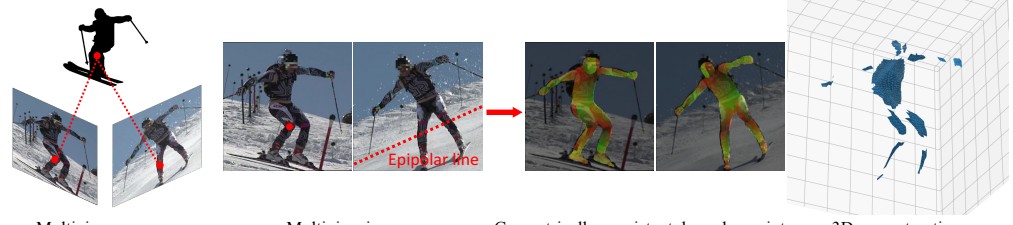

| Multiview camera | Multiview images | Geometrically consistent dense keypoints | 3D reconstruction |

Figure 1: We use unlabeled multiview images to learn a dense keypoint model via the epipolar geometry in an end-to-end fashion. As a byproduct, we can reconstruct the 3D body surface by triangulating visible regions of body parts.

new semi-supervised method to learn a dense keypoint detection model from the unlabeled multiview images via the epipolar constraint as shown in Figure 1.

Our main conjecture is that the dense keypoint model is optimal only if it is geometrically consistent across views. That is, every pair of corresponding keypoints, independently predicted by two views, must satisfy the epipolar constraint [9]. However, enforcing the epipolar constraint to learn a dense keypoint model is challenging because (1) the ground truth 3D model is unknown and thus the projections of the 3D model cannot be used as the ground truth dense keypoints; (2) the predicted dense keypoints are inaccurate and continuous over the body surface, and therefore, existing multiview supervision approaches [51, 38, 45] for sparse keypoints are not applicable. In these previous methods, the epipolar constraint was enforced between two keypoints (or features) of which semantic meaning was explicitly defined by a finite set of joints (e.g., elbow channel in a network); and (3) establishing correspondences across views requires knowing an inverse mapping from the body surface to the image that can be neither analytically derived nor differentiable. These challenges limit the performance of previous work [22] that relies on iterative offline bootstrapping, which is not end-to-end trainable, or requires additional parameters to learn for 3D reconstruction[1].

We tackle these challenges through a *probabilistic epipolar constraint* by incorporating an uncertainty in correspondences. This new constraint encodes the two desired properties. (1) Soft correspondence: given a keypoint in one image, we define matchability—the likelihood of correspondence for all predicted keypoints in another image based on the distance in the canonical body surface coordinate (e.g., texture coordinate). This allows evaluating geometric consistency in the form of a weighted average of epipolar errors over continuous body surface coordinates, eliminating the requirement of exact correspondences. (2) Geometric consistency: we generalize symmetric Sampson distance [9] for all possible pairs of keypoints from two views to enforce the epipolar constraint, collectively. With these properties, we derive a new differentiable multiview consistency measure that is label-agnostic, allowing us to utilize a large number of the unlabeled multiview images without explicit 3D reconstruction.

We design an end-to-end trainable twin network architecture that takes a pair of images as an input and outputs geometrically consistent dense keypoint fields. This network design builds the affinity maps between two keypoint fields based on the matchability and epipolar errors, which facilitates measuring the probabilistic epipolar errors for all possible correspondences. In addition, inspired by knowledge distillation, we use a pretrained model to regularize network learning, which can prevent degenerate cases. Our method shows superior performance compared to existing methods, including non-differentiable bootstrapping [22] in terms of keypoint accuracy, multiview consistency, and 3D reconstruction accuracy.

Our contributions include: (1) a novel formulation of probabilistic epipolar constraint that can be used to enforce multiview consistency on continuous dense keypoint fields in a differentiable way; (2) a new design of the neural network that enable to precisely measure the probabilistic epipolar error, which allows utilizing a large number of the unlabeled multiview images; (3) a distillation-based regularization to prevent degenerate model learning; (4) strong performance on real-world multiview image data, including Human3.6M [11], Ski-Pose [40], and OpenMonkeyPose [3], outperforming existing methods including non-differentiable dense keypoint learning [22].

**Broader Impact Statement** The ability to understand animals' individual and social behaviors is of central importance to multiple disciplines such as biology, neuroscience, and behavioral science.

---

[1]An analogous insight has been used for fundamental matrix, directly computed from correspondences that does not require additional variables for 3D reconstruction.

Measuring their behaviors has been extremely challenge due to limited annotated data. This approach offers a way to address this challenge with a limited number of annotated data, which will lead to a scalable behavioral analysis. The negative societal impact of this work is minimum.

## 2 Related Work

Our framework aims at training a dense keypoint field estimation model via multiview supervision. We briefly review the related works.

**Dense Keypoint Field Estimation** Finding dense correspondence fields between two images is a challenging problem in computer vision. 3D measurements (e.g., depth and pointcloud) can provide a strong geometric cues which enable matching of deformable shapes [42, 29, 48]. Similarly, in 2D, visual and geometric cues have been used to find the dense correspondence fields in an unsupervised learning [56, 6, 4, 44, 43]. Notably, for special foreground targets such as humans, the dense matching problem can be cast as finding a dense keypoint field that maps pixel coordinates to a canonical body surface coordinates [24] (e.g., DensePose [8]). These works were built upon a large amount of data labeled by crowd-workers and generalized to learn the correspondence fields for face [2] and chimpanzees [36]. Key limitations of these approaches are the inaccuracy of labeling and requirement of large labeled data. We address these limitations by formulating multiview supervision that can enforce geometric consistency, which allows utilizing a large amount of unlabeled multiview images.

**Multiview Feature Learning** Epipolar geometry can be used to learn a visual representation by transferring visual information from one image to another via epipolar lines or 3D reconstruction. For instance, a fusion layer can be learned to fuse feature maps across views [30]. Such fusion models can be factored into shape and camera components to reduce the number of learnable parameters and improve generalizability [49]. Given the camera calibration, a light fusion module can be learned to directly fuse deep features [10] or heatmaps [55] from other view along corresponding epipolar line. Several works combine multiview image features to form 3D features [13, 46] or view-invariant feature in 2D [32].

**Multiview Supervision** Synchronized multiview images [11, 15, 52] possess a unique geometric property: images are visually similar yet geometrically distinctive, provided by stereo parallax. Such property offers a new opportunity to learn a geometrically coherent representation without labels. Bootstrapping by 3D reconstruction [22] can be used to learn a keypoint detector supervised by the projection of the 3D reconstruction to enforce cross-view consistency. MONET [51] enables an end-to-end learning by eliminating the necessity of 3D reconstruction and directly minimizing epipolar error. Learning keypoints can be combined with 3D pose estimation [34, 12] by enforcing predicting the same pose in all views while using a few labeled examples with 3D or 2D pose annotations to prevent degeneration. One can alleviate the need for large amounts of annotations by matching the predicted 3D pose with the triangulated pose [19]. Several works further learn a latent representation encoding 3D geometry from images [5, 25] or 2D pose [33, 41] by enforcing consistent embedding, texture [28], and view synthesis [45] across views. Unlike these approaches designed for sparse keypoints where the geometric consistency is applied on finite points, we study geometric consistency on continuous dense keypoint fields. Capture Dense [22] is the closest work to ours, which uses bootstrapping through 3D reconstruction of human mesh model [16]. However, due to the non-differentiable nature of bootstrapping, it is not end-to-end trainable. A temporal consistency has been also used for self-supervise the dense keypoint detector [27].

## 3 Method

We present a novel method to learn a dense keypoint detector by using unlabeled multiview images. We formulate the epipolar constraint for continuous correspondence fields, which allows us to enforce geometric consistency between views.

### 3.1 Dense Epipolar Geometry

Given a pair of corresponding images from two different views, a pair of corresponding points $\mathbf{x} \leftrightarrow \mathbf{x}'$, are related by a fundamental matrix given the calibrated cameras, i.e. $\widetilde{\mathbf{x}}'^{\mathsf{T}} \mathbf{F} \widetilde{\mathbf{x}} = 0$, where $\mathbf{F} \in \mathbb{R}^{3 \times 3}$ is the fundamental matrix, and $\widetilde{\mathbf{x}} \in \mathbb{P}^2$ is a homogeneous representation of $\mathbf{x}$. The

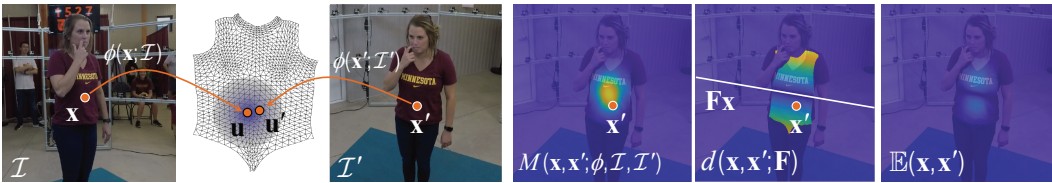

Figure 2: A dense keypoint field maps a point in an image to the canonical body surface coordinate, i.e., $\mathbf{u} = \phi(\mathbf{x}; \mathcal{I})$. Establishing a correspondence between two view images requires the analytic inverse of $\phi$ which does not exist in general. We present a matchability $M(\mathbf{x}, \mathbf{x}'; \phi, \mathcal{I}, \mathcal{I}')$, a likelihood of matching through the body surface coordinate. We combine the matchability with the epipolar error $d(\mathbf{x}, \mathbf{x}'; \mathbf{F})$ to obtain a probabilistic epipolar error $\mathbb{E}(\mathbf{x}, \mathbf{x}')$.

measure of geometric consistency between the two images can be written as [9]:

$$d(\mathbf{x}, \mathbf{x}'; \mathbf{F}) = \frac{|\widetilde{\mathbf{x}}'^{\mathsf{T}} \mathbf{F} \widetilde{\mathbf{x}}|}{\sqrt{(\mathbf{F}\widetilde{\mathbf{x}})_1^2 + (\mathbf{F}\widetilde{\mathbf{x}})_2^2}}, \quad \mathbf{x} \leftrightarrow \mathbf{x}', \tag{1}$$

where $(\mathbf{F}\mathbf{x})_i$ is the $i^{\text{th}}$ entry of $\mathbf{F}\mathbf{x}$. This measures epipolar error that is the Euclidean distance between $\mathbf{x}'$ and the epipolar line $\mathbf{F}\widetilde{\mathbf{x}}$.

Consider an injective dense keypoint mapping $\phi : \mathbb{R}^2 \rightarrow \mathbb{R}^2$ that maps a pixel coordinate to a canonical 2D body surface coordinate, i.e., $\mathbf{u} = \phi(\mathbf{x}; \mathcal{I})$ where $\mathbf{x} \in \Theta(\mathcal{I})$ is the set of the foreground pixels in the image $\mathcal{I}$, and $\mathbf{u} \in \mathbb{R}^2$ is the 2D coordinate in the body surface as shown in Figure 2. This dense keypoint mapping can be learned from annotations, e.g., DensePose [8] for humans where it maps to the texture coordinate of 3D human body surfaces. This mapping is incomplete because there exists missing correspondences in an image due to occlusion. One can find a correspondence between the two images through $\mathbf{u}$, i.e., $\phi^{-1}(\mathbf{u}; \mathcal{I}) \leftrightarrow \phi^{-1}(\mathbf{u}; \mathcal{I}')$ where $\mathcal{I}$ and $\mathcal{I}'$ the two view images. However, $\phi$ is an injective mapping where the analytic inverse does not exist in general.

Given a point $\mathbf{x}$ in the image $\mathcal{I}$, one can measure the expectation of geometric error by a nearest neighbor search on the body surface space:

$$\mathbb{E}(\mathbf{x}) = d(\mathbf{x}, \mathbf{x}'; \mathbf{F}), \quad \mathbf{x}' = \operatorname*{argmin}_{\mathbf{x}' \in \Theta(\mathcal{I}')} \|\phi(\mathbf{x}; \mathcal{I}) - \phi(\mathbf{x}'; \mathcal{I}')\|, \tag{2}$$

where $\mathbb{E}(\mathbf{x})$ is the expectation of geometric error at $\mathbf{x}$. The expectation of the geometric error measures the epipolar error over all possible matches in the other view image, i.e., $\forall \mathbf{x}' \in \Theta(\mathcal{I}')$.

There are two limitations in the nearest neighbor search: (1) The correspondences are not exact, leading to a biased estimate of the geometric error expectation. (2) The argmin operation is not differentiable, which cannot be used to learn a dense keypoint detector in an end-to-end fashion.

Instead, we address these limitations by making use of a soft correspondence, or *matchability*—a likelihood of a point being matched to another point as shown in Figure 2:

$$M(\mathbf{x}, \mathbf{x}'; \phi, \mathcal{I}, \mathcal{I}') = P\left(\phi(\mathbf{x}; \mathcal{I}), \phi(\mathbf{x}'; \mathcal{I}'); \{\phi(\mathbf{y}; \mathcal{I}')\}_{\mathbf{y} \in \Theta(\mathcal{I}')}\right), \tag{3}$$

where $P(\mathbf{u}, \mathbf{u}'; \Omega)$ is the probability of matching between $\mathbf{u}$ and $\mathbf{u}'$:

$$P(\mathbf{u}, \mathbf{u}'; \Omega) = \exp\left(-\frac{\|\mathbf{u} - \mathbf{u}'\|^2}{2\sigma^2}\right) \Big/ \sum_{\mathbf{v} \in \Omega} \exp\left(-\frac{\|\mathbf{u} - \mathbf{v}\|^2}{2\sigma^2}\right). \tag{4}$$

$\Omega$ is the domain of the canonical coordinates, $\sigma$ is the standard deviation that controls the smoothness of matching, e.g., when $\sigma \rightarrow 0$, it approximates the nearest neighbor search.

We use the matchability to form a probabilistic epipolar error:

$$\mathbb{E}(\mathbf{x}, \mathbf{x}') = M(\mathbf{x}, \mathbf{x}'; \phi, \mathcal{I}, \mathcal{I}') d(\mathbf{x}, \mathbf{x}'; \mathbf{F}), \tag{5}$$

where $\mathbb{E}(\mathbf{x}, \mathbf{x}')$ is the epipolar error between x and x' weighted by the matchability.

The error expectation of $\mathbf{x}$ can be computed by marginalizing over all $\mathbf{x}'$:

$$\mathbb{E}(\mathbf{x}) = \sum_{\mathbf{x}' \in \Theta(\mathcal{I}')} \mathbb{E}(\mathbf{x}, \mathbf{x}'). \tag{6}$$

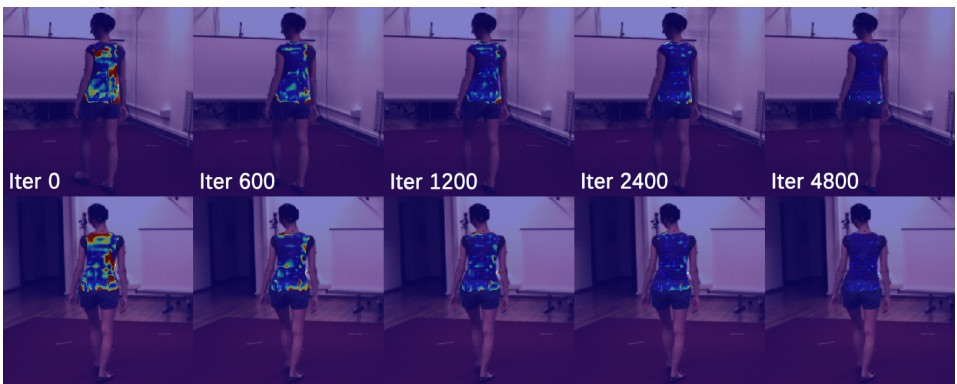

Figure 3: Our multiview supervision progressively minimizes the epipolar error between two views (top and bottom) as learning the dense keypoint detection model. The keypoint detection, independently by a pretrained model (Iter 0), is not geometrically consistent. As the optimization progresses, the error is significantly reduced, resulting in a geometrically coherent model.

That is, the expectation of the geometric error of $\mathbf{x}$ measures a weighted average of epipolar errors for all possible correspondences. The higher matchability, the more contribution to the error expectation. Unlike Equation (2), Equation (6) does not include the non-differentiable argmin operation. Further, it takes into account all possible pairs of correspondences between two images, which eliminates the bias introduced by the nearest neighbor search.

Given the error expectations over the dense keypoint field, we derive a symmetric dense epipolar error that measures geometric consistency between two images:

$$\mathbb{E}(\mathcal{I}, \mathcal{I}') = \frac{1}{V} \sum_{\mathbf{x} \in \Theta(\mathcal{I})} v(\mathbf{x}; \phi, \mathcal{I}, \mathcal{I}') \mathbb{E}(\mathbf{x}) + \frac{1}{V'} \sum_{\mathbf{x}' \in \Theta(\mathcal{I}')} v(\mathbf{x}'; \phi, \mathcal{I}, \mathcal{I}') \mathbb{E}(\mathbf{x}'), \tag{7}$$

where $v(\mathbf{x}; \phi, \mathcal{I}, \mathcal{I}') \in \{0, 1\}$ is a visibility indicator:

$$v(\mathbf{x}; \phi, \mathcal{I}, \mathcal{I}') = \begin{cases} 1, & \exists \mathbf{x}' \in \Theta(\mathcal{I}'), \|\phi(\mathbf{x}; \mathcal{I}) - \phi(\mathbf{x}'; \mathcal{I}')\| < \epsilon \\ 0, & \text{otherwise.} \end{cases} \tag{8}$$

$V = \sum_{\mathbf{x} \in \Theta(\mathcal{I})} v(\mathbf{x}; \phi, \mathcal{I}, \mathcal{I}')$ and $V' = \sum_{\mathbf{x}' \in \Theta(\mathcal{I}')} v(\mathbf{x}'; \phi, \mathcal{I}, \mathcal{I}')$ are the numbers of foreground pixels visible in $\mathcal{I}'$ and $\mathcal{I}$, respectively. $\epsilon$ is a correspondence tolerance defined in the canonical surface coordinate.

In fact, this dense epipolar error is a generalization of Sampson distance [9] that defines epipolar error between explicit point correspondences (point-to-point). We generalize it to model probabilistic correspondence between two set of points (field-to-field).

Existing approaches such as bootstrapping [22] establish the matching through 3D reconstruction of mesh and enforce the geometric error in an alternating fashion due to the non-differentiability of matching. Our differentiable formulation allows learning the dense keypoint detector in an end-to-end manner, which is flexible and shows superior performance.

### 3.2 Multiview Semi-supervised Learning

We learn the dense keypoint detector $\phi$ by minimizing the following error:

$$\mathcal{L} = \lambda_{\mathrm{L}} \mathcal{L}_{\mathrm{L}} + \lambda_{\mathrm{M}} \mathcal{L}_{\mathrm{M}} + \lambda_{\mathrm{R}} \mathcal{L}_{\mathrm{R}} + \lambda_{\mathrm{T}} \mathcal{L}_{\mathrm{T}}, \tag{9}$$

where $\mathcal{L}_{\mathrm{L}}$ is the labeled data loss, $\mathcal{L}_{\mathrm{M}}$ is the multiview geometric consistency loss, $\mathcal{L}_{\mathrm{R}}$ is the regularization loss, and $\mathcal{L}_{\mathrm{T}}$ is the multiview photometric consistency loss. $\lambda_{\mathrm{L}}$, $\lambda_{\mathrm{M}}$, $\lambda_{\mathrm{R}}$ and $\lambda_{\mathrm{T}}$ are the weights to control their relative importance.

**Supervised Loss** We learn the dense keypoint detector from the labeled dataset $\mathcal{D}_{\mathrm{L}}$ by minimizing the following error:

$$\mathcal{L}_{\mathrm{L}} = \sum_{\{\mathcal{I}, \mathbf{U}\} \in \mathcal{D}_{\mathrm{L}}} \sum_{\mathbf{x} \in \Theta(\mathcal{I})} \|\mathbf{U}_{\mathbf{x}} - \phi(\mathbf{x}; \mathcal{I})\|_1 \tag{10}$$

where $\mathbf{U} \in \mathbb{R}^{2 \times H \times W}$ is the ground truth canonical surface coordinates for dense keypoints, $\mathbf{U_x}$ is the ground truth canonical surface coordinate at $\mathbf{x}$.

**Multiview Geometric Consistency Loss** We learn the dense keypoint detector from the unlabeled multiview dataset $\mathcal{D}_{\mathrm{U}}$ by minimizing dense epipolar error over image pairs:

$$\mathcal{L}_{\mathrm{M}} = \sum_{\{\mathcal{I}, \mathcal{I}'\} \in \mathcal{D}_{\mathrm{U}}} \mathbb{E}(\mathcal{I}, \mathcal{I}'), \tag{11}$$

where $\mathbb{E}(\mathcal{I}, \mathcal{I}')$ is the dense epipolar error between two corresponding images $\mathcal{I}$ and $\mathcal{I}'$, defined in Equation (14). This loss progressively minimizes the dense epipolar error between image pair as learning the dense keypoint detection model as shown in Figure 3.

**Distillation Based Regularization Loss** Enforcing multiview consistency alone can lead to degenerate cases. For instance, consider a linear transformation in the body surface, e.g., $\mathbf{v} = \mathbf{Tu}$ where $\mathbf{T} \in \mathbb{R}^{2 \times 2}$ is a similarity transformation. Any $\phi$ that satisfies the following condition can be equivalent dense keypoint detector:

$$\phi(\mathbf{x}; \mathcal{I}) \equiv \mathbf{T}\phi(\mathbf{x}; \mathcal{I}). \tag{12}$$

This indicates that there exist an infinite number of dense keypoint detectors that satisfy the epipolar geometry. To alleviate this geometric ambiguity, we use a distillation-based regularization using a pretrained model. Let $\phi_0$ be a dense keypoint detector pretrained by the labeled data. We prevent the learned detector $\phi$ from deviating too much from the pretrained detector $\phi_0$ for the unlabeled data:

$$\mathcal{L}_{\mathrm{R}} = \sum_{\mathcal{I} \in \mathcal{D}_{\mathrm{U}}} \sum_{\mathbf{x} \in \Theta(\mathcal{I})} \|\phi_0(\mathbf{x}; \mathcal{I}) - \phi(\mathbf{x}; \mathcal{I})\|_1, \tag{13}$$

where $\mathcal{L}_{\mathrm{R}}$ is the loss for the distillation-based regularization, minimizing the difference from the pretrained model.

**Multiview Photometric Consistency Loss** We leverage photometric consistency across views. Assuming ambient light, pixels across views corresponding to the same 3D point in space should have the same RGB value. Similar to dense epipolar error $\mathbb{E}(\mathcal{I}, \mathcal{I}')$, we define dense photometric error $\mathbb{T}(\mathcal{I}, \mathcal{I}')$ as:

$$\mathbb{T}(\mathcal{I}, \mathcal{I}') = \frac{1}{V} \sum_{\mathbf{x} \in \Theta(\mathcal{I})} v(\mathbf{x}; \phi, \mathcal{I}, \mathcal{I}') \mathbb{T}(\mathbf{x}) + \frac{1}{V'} \sum_{\mathbf{x}' \in \Theta(\mathcal{I}')} v(\mathbf{x}'; \phi, \mathcal{I}, \mathcal{I}') \mathbb{T}(\mathbf{x}'), \tag{14}$$

where $\mathbb{T}(\mathbf{x})$ is the expectation of photometric error of $\mathbf{x}$ similar to $\mathbb{E}(\mathbf{x})$, i.e. $\mathbb{T}(\mathbf{x}) = \sum_{\mathbf{x}' \in \Theta(\mathcal{I}')} \mathbb{T}(\mathbf{x}, \mathbf{x}')$ where $\mathbb{T}(\mathbf{x}, \mathbf{x}') = M(\mathbf{x}, \mathbf{x}'; \phi, \mathcal{I}, \mathcal{I}')\|\mathcal{I}(\mathbf{x}) - \mathcal{I}'(\mathbf{x}')\|^2$.

Then we compute multiview photometric consistency loss as:

$$\mathcal{L}_{\mathrm{T}} = \sum_{\{\mathcal{I}, \mathcal{I}'\} \in \mathcal{D}_{\mathrm{U}}} \mathbb{T}(\mathcal{I}, \mathcal{I}'), \tag{15}$$

### 3.3 Network Design

We design a new network architecture composed of twin networks to learn the dense keypoint detector by enforcing multiview consistency over dense keypoint fields as shown in Figure 4. Each network is made of a fully convolutional network that outputs the dense keypoint field per body part.

Given two dense keypoint fields, we compute the probabilistic epipolar error by constructing two affinity matrices: matchability matrix and epipolar matrix, $\mathbf{M}, \mathbf{E} \in \mathbb{R}^{|\Theta(\mathcal{I})| \times |\Theta(\mathcal{I}')|}$ where $|\Theta(\mathcal{I})|$ is the cardinality of the range of foreground pixels. These two matrices are defined by:

$$\mathbf{M}_{ij} = M(\mathbf{x}_i, \mathbf{x}_j; \phi, \mathcal{I}, \mathcal{I}'), \quad \mathbf{E}_{ij} = d(\mathbf{x}_i, \mathbf{x}_j; \mathbf{F}) \tag{16}$$

where $\mathbf{M}_{ij}$ is the $i, j$ entry of the matrix $\mathbf{M}$. $\mathbf{x}_i$ and $\mathbf{x}_j$ are the $i^{\mathrm{th}}$ and $j^{\mathrm{th}}$ foreground pixels from the images $\mathcal{I}$ and $\mathcal{I}'$, respectively. We also compute the visibility maps $\mathbf{V}_i = v(\mathbf{x}; \phi, \mathcal{I}, \mathcal{I}')$ and $\mathbf{V}'_i = v(\mathbf{x}'; \phi, \mathcal{I}, \mathcal{I}')$.

We design a new operation to measure the dense epipolar error in Equation (14):

$$\mathbb{E}(\mathcal{I}, \mathcal{I}') = \frac{1}{V} \mathbf{V}^{\mathsf{T}} (\mathbf{M} \odot \mathbf{E}) \mathbf{1}_{|\Theta(\mathcal{I}')|} + \frac{1}{V'} \mathbf{V}'^{\mathsf{T}} (\mathbf{M} \odot \mathbf{E})^{\mathsf{T}} \mathbf{1}_{|\Theta(\mathcal{I})|}, \tag{17}$$

where $\mathbf{1}_n$ is the $n$-dimensional vector of which entries are all one. $\odot$ is the element-wise multiplication of matrices. Note that dense epipolar error $\mathbb{T}(\mathcal{I}, \mathcal{I}')$ can be computed following the same operations.

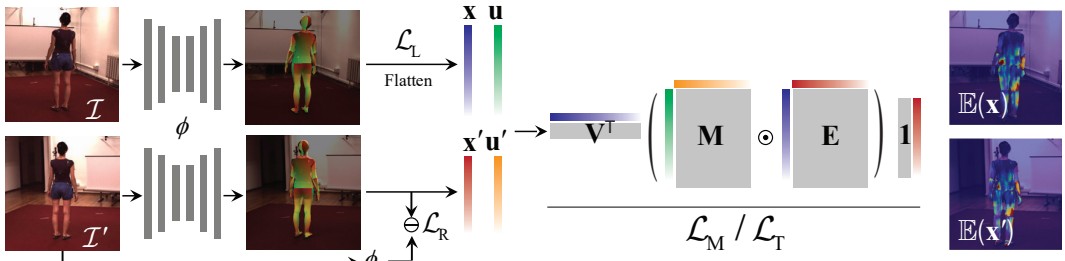

Figure 4: We design a new architecture composed of twin networks that detect dense keypoint fields. The dense keypoint fields from two views are combined to form two affinity matrices: matchability $\mathbf{M}$ and epipolar error $\mathbf{E}$. $\mathbf{M}$ is obtained from the dense keypoint fields ($\mathbf{u}$ and $\mathbf{u}'$), and $\mathbf{E}$ is obtained from the epipolar error of pixel coordinates ($\mathbf{x}$ and $\mathbf{x}'$). These matrices allow us to compute dense epipolar errors and subsequent multiview geometric consistency loss $\mathcal{L}_{\mathrm{M}}$. Same operations are applied to compute $\mathcal{L}_{\mathrm{T}}$. In addition, we make use of distillation-based regularization using a pretrained model $\phi_0$ to avoid degenerate cases ($\mathcal{L}_{\mathrm{R}}$). We measure the labeled loss $\mathcal{L}_{\mathrm{L}}$ if the ground truth dense keypoint field is available. $\odot$ is the element-wise multiplication of matrices. $\ominus$ is the minus operation between dense keypoint predictions.

## 4 Experiments

We perform experiments on human and monkey targets as two example applications to evaluate the effectiveness of our proposed semi-supervised learning pipeline.

### 4.1 Implementation Details

We use HRNet [18] as the backbone network followed by four head networks made up of convolutional layers to predict foreground mask, body part index, and UV coordinates on the canonical body surface, respectively. Each network takes as an input a $224 \times 224$ image and outputs 15-channel (for foreground mask head only) or 25-channel $56 \times 56$ feature maps [8].

We train the network in two stages. In the first stage, we train an initial model using the labeled data by the labeled data loss $\mathcal{L}_{\mathrm{L}}$. Specifically, for the human dense keypoints, we use 48K human instances in DensePose-COCO [8] training set to train the initial model. For the monkey dense keypoints, we directly use the pretrained model for chimpanzees [36] as the initial model since we do not have an access to the labeled data. In the second stage, we train our model on unlabeled multiview data leveraging all losses. The initial model is used for two purposes: (1) the pre-trained model $\phi_0$ with weights fixed for distillation-based regularization and (2) the initial model to be refined via multiview supervision.

### 4.2 Evaluation Datasets

**Human3.6M** [11] is a large-scale indoor multiview dataset captured by 4 cameras for 3D human pose estimation. It contains 3.6 millions of images captured from 7 subjects performing 15 different daily activities, e.g. Walking, Greeting, and Discussion. Following common protocols, we use subject S1, S5, S6, S7 and S8 for training, and reserve subject S9 and S11 for testing. Following [54] and [53], we leverage SMPL [24] parameters generated by HMR [17] via applying MoSh [23] to the sparse 3D MoCap marker data to recover ground truth 3D human meshes. We further perform Procrustes analysis [39] to align them with ground truth 3D poses in global coordinate system and then render ground truth IUV maps using PyTorch3D [31]. Since it is the only multiview human dataset that we have access to ground truth 3D mesh / IUV maps, we use this dataset to perform comprehensive experiment and studies.

**3DPW** [47] is an in-the-wild dataset made of single view images with accurate 3D human pose and shape annotations. We use the test split of this dataset for dense keypoint accuracy evaluations (3D metrics do not apply). Ground truth IUV maps are rendered from 3D pose and shape annotations by leveraging the provided camera parameters.

**Ski-Pose PTZ-Camera Dataset** [40] is a multiview dataset capturing competitive skiers performing giant slalom runs. 6 synchronized and calibrated pant-tile-zoom-cameras (PTZ) cameras are used to track a single skier at a time. The global locations of the cameras were measured by a tachymeter

theodolite. It contains 8.5K training images and 1.7K testing images. We use this dataset to evaluate generalization towards in-the-wild multiview settings. We use its standard train/test split to train and evaluation our model. We select 6 adjacent view pairs to form training samples.

**OpenMonkeyPose** [3] is a large landmark dataset of rhesus macaques captured by 62 synchronized multiview cameras. It consists of nearly 200K labeled images with four macaque subjects that freely move in a large cage while performing foraging tasks. Each monkey instance is annotated with 13 2D and 3D joints. We use this dataset to show our model's ability on transferring dense keypoints to monkey data. We split about 64K images for training and 12K images for testing. For training, we generate the densepose of monkey data using a pretrained model [36] as pseudo-labels. These pseudo-labels are then used for refining the pretrained model.

### 4.3 Baselines

In our experiments, we consider four baselines: (1) A model fully-supervised by labeled data, i.e. DensePose-COCO [8], which is also our initial model for semi-supervised learning. We refer to this as Supervised in the following. (2) A model trained using multiview bootstrap strategy [38, 22], where multiview triangulation results from previous stage are used as the pseudo ground truth. (3) A learnable 3D mesh estimation method where dense keypoint estimation can be acquired by reprojecting 3D mesh to image domain, e.g. HMR [17]. (4) A 3D mesh estimation framework similar to HMR but additionally incorporating Model-fitting in the Loop, e.g. SPIN [20]. Supervised is also used as an initial model for our approach.

### 4.4 Metrics

We evaluate the performance of our dense keypoint model using metrics from three aspects: (1) geometric consistency, (2) accuracy of dense keypoints, and (3) accuracy of 3D reconstruction from multiple views.

**Geometric Consistency** We use epipolar distance (unit: pixel) averaged over views and frames as the metric for evaluating multiview geometric consistency. Ideally, two dense keypoints corresponding to the same point on 3D surface should have epipolar distance equal to 0. This metric can be evaluated on any multiview dataset with ground truth camera parameters available.

**Dense Keypoint Accuracy** We evaluate the model's performance on dense keypoint accuracy from two aspects: (1) Ratio of Correct Point (RCP) and (2) Ratio of Correct Instances (RCI). RCP evaluates correspondence accuracy over the whole image domain. Specifically, it records the ratio of foreground pixels on images with corresponding 3D body surface correctly predicted as a function of geodesic distance threshold, where the prediction is considered correct if its geodesic distance to the ground truth is below the threshold (10cm and 30cm). RCIs consider instance-wise accuracy where an instance is declared to be correct if its geodesic point similarity (GPS) [8] is above the threshold. We also report the mean RCI (mRCI) and the mean GPS for all instances (mGPS).

**Reconstruction Accuracy** Given dense keypoints, we measure 3D reconstruction error by triangulating them in 3D. We compute Mean Per mesh Vertex Position Error (MPVPE) as the metric for reconstruction accuracy, which is defined as the mean euclidean distance between triangulated vertices and corresponding ground truth ones. In addition, inspired by geodesic point similarity [8], we define vertex similarity as: $\text{VS} = \frac{1}{|V|} \sum_{\mathbf{v}_i \in V} \exp\left(-\frac{-d(\hat{\mathbf{v}}_i - \mathbf{v}_i)^2}{2\kappa^2}\right)$, where $d(\hat{\mathbf{v}}_i - \mathbf{v}_i)$ is the euclidean distance between triangulated vertex $\hat{\mathbf{v}}_i$ and corresponding ground truth one $\mathbf{v}_i$, and $V$ is the set of visible ground truth vertices from both views. $\kappa$ is a normalizing parameter. For $\mathbf{v}_i$ that does not correspond to the triangulated vertex, it is set to infinity. Further, to account for false positives in triangulated vertices (vertices not visible from both views), we define masked vertex similarity (MVS) as $\text{MVS} = \sqrt{\text{VS} \cdot I}$, where $I$ is the intersection over union between the set of triangulated vertices and $V$. We report mean MVS (mMVS) over all instances.

### 4.5 Evaluation on Human3.6M Dataset

We use Human3.6M dataset to perform (1) comprehensive cross-method evaluations, (2) study on model's generalizability towards new views, and (3) ablation study on losses used for training. Results are summarized in Table 1 with all metrics reported.

**Cross-method Evaluation** We evaluate the performance of our model against other methods as shown in the comparison block in Table 1 and show qualitative results (1st column in Figure 5). For the metrics of keypoint accuracy and geometry consistency, ours outperforms all other methods

| | Method | Keypoint accuracy | | | | Geom. consistency | Recon. accuracy | |
|---|---|---|---|---|---|---|---|---|
| | | $AUC_{10}$ | $AUC_{30}$ | mRCI | mGPS | Epi. error | MPVPE | mMVS |
| **Comparison** | Supervised | 0.445 | 0.729 | 0.761 | 0.856 | 6.09 | 60.04 | 0.498 |
| | Bootstrapping [22] | 0.454 | 0.732 | 0.763 | 0.857 | 5.90 | 58.56 | 0.438 |
| | HMR [17] | **0.513** | 0.701 | 0.610 | 0.780 | 3.67 | 50.10 | **0.831** |
| | SPIN [20] | 0.472 | 0.633 | 0.459 | 0.704 | 3.32 | **50.46** | 0.725 |
| | Ours | 0.486 | **0.745** | **0.770** | **0.861** | **2.05** | 53.04 | 0.561 |
| **General.** | Supervised (Test view 1,3) | 0.428 | 0.724 | 0.760 | 0.855 | 5.87 | 58.98 | 0.521 |
| | Ours (Train view 0,2 / Test view 1,3) | **0.454** | **0.734** | **0.766** | **0.858** | **3.62** | **55.38** | **0.555** |
| | Supervised (Test view 0,2) | 0.460 | 0.735 | 0.762 | 0.856 | 5.83 | 57.90 | 0.462 |
| | Ours (Train view 1,3 / Test view 0,2) | **0.519** | **0.756** | **0.771** | **0.861** | **3.04** | **49.94** | **0.536** |
| **Ablation** | Supervised ($\mathcal{L}_L$) | 0.445 | 0.729 | 0.761 | 0.856 | 6.09 | 60.04 | 0.498 |
| | $\mathcal{L}_L + \mathcal{L}_M$ | 0.113 | 0.438 | 0.472 | 0.710 | 1.86 | 167.61 | 0.330 |
| | $\mathcal{L}_L + \mathcal{L}_T$ | 0.164 | 0.519 | 0.569 | 0.759 | 6.03 | 128.26 | 0.510 |
| | $\mathcal{L}_L + \mathcal{L}_M + \mathcal{L}_T$ | 0.126 | 0.471 | 0.509 | 0.730 | 1.97 | 138.43 | 0.409 |
| | $\mathcal{L}_L + \mathcal{L}_R$ | 0.446 | 0.730 | 0.762 | 0.857 | 5.94 | 59.64 | 0.492 |
| | $\mathcal{L}_L + \mathcal{L}_R + \mathcal{L}_M$ | 0.475 | 0.741 | 0.768 | 0.860 | 2.06 | 57.92 | 0.535 |
| | $\mathcal{L}_L + \mathcal{L}_R + \mathcal{L}_T$ | 0.458 | 0.734 | 0.764 | 0.858 | 5.13 | 54.88 | 0.540 |
| | $\mathcal{L}_L + \mathcal{L}_R + \mathcal{L}_M + \mathcal{L}_T$ | **0.486** | **0.745** | **0.770** | **0.861** | **2.05** | **53.04** | **0.561** |

Table 1: We performance cross-method evaluation, study on model's generalizability towards new views and ablation study on Human3.6M dataset and report performance on keypoint accuracy, geometric consistency and reconstruction accuracy. Note that HMR and SPIN are trained with the 3D ground truth (pre-estimated mesh) of Human3.6M while other algorithms including ours predict dense keypoints without the 3D ground truth. (Epipolar error unit: pixel; MPVPE unit: mm)

(except for $AUC_{10}$ only second to HMR [17]). Note that HMR and SPIN are trained with the 3D ground truth (pre-estimated mesh) of Human3.6M while other algorithms including ours predict dense keypoints without the 3D ground truth. Having the 3D ground truth is a significant advantage while it requires a substantial additional amount of 3D annotation effort. It is expected to outperform the ones without it, in particular, on reconstruction accuracy metrics. We included these baselines to provide the upper bound of our semi-supervised method as a reference. Note that despite the lack of the 3D ground truth, ours still outperforms HMR and SPIN by a margin of up to 26.2% and 67.8% on keypoint accuracy metrics on three keypoint accuracy metrics. Compared to "Supervised" and "Bootstrapping" which are close to our setup, ours outperform on all metrics by a margin of up to 9.2% and 7.0% on keypoint accuracy metrics.

**Generalizability towards new views** We evaluate generalization by testing on different views: two views are used for training and other two views are used for testing. The results are summarized in the generalization block in Table 1. As can be seen, although a model trained purely on one camera pair does not use any sample captured by the other pair, its performance still improves on all metrics on top of Supervised model by a margin of 0.6%-12.8% on keypoint accuracy, 38.3% - 47.9% on geometric consistency, and 6.5% - 13.7% on reconstruction accuracy. This shows that model trained by our semi-supervised approach can be generalized to new views.

**Ablation Study** We conduct an ablation study to evaluate the impact of each loss. The results are reported in the ablation block in Table 1. Note that we propose semi-supervised learning thus $\mathcal{L}_L$ is always used. The inferior performance of model trained with $\mathcal{L}_L + \mathcal{L}_M$ or $\mathcal{L}_L + \mathcal{L}_T$ or $\mathcal{L}_L + \mathcal{L}_M + \mathcal{L}_T$ proves that refining initial model by multiview loss alone suffers from degeneration (Section 3.2). This limitation can be addressed by adding the regularization $\mathcal{L}_R$. Note that regularization itself does not add any value: it is identical to the supervised model in theory. This is empirically verified since $\mathcal{L}_L + \mathcal{L}_R$ performs similarly to $\mathcal{L}_L$. Both multiview consistency losses show effectiveness: $\mathcal{L}_L + \mathcal{L}_R + \mathcal{L}_M$ and $\mathcal{L}_L + \mathcal{L}_R + \mathcal{L}_T$ outperform $\mathcal{L}_L + \mathcal{L}_R$ on all metrics by a margin up to 6.5% / 2.7% for keypoint accuracy metrics and 8.7% / 9.8% for reconstruction accuracy metrics respectively. $\mathcal{L}_L + \mathcal{L}_R + \mathcal{L}_M + \mathcal{L}_T$ further achieves better results on all metrics. Multiview geometric consistency loss mainly contributes to the improvement on keypoint accuracy and geometric consistency metrics, while multiview photometric consistency loss mainly contributes to reconstruction accruacy metrics.

### 4.6 Evalution on 3DPW Dataset

To further validate the usefulness of our method in terms of dense keypoint accuracy, in addition to detection accuracy on multiview image data, we conduct another cross-method evaluation on 3DPW dataset (single-view dense keypoint detection) on the keypoint accuracy met-

| | Method | Keypoint accuracy | | | |
|---|---|---|---|---|---|
| | | $AUC_{10}$ | $AUC_{30}$ | mRCI | mGPS |
| **Comparison** | Supervised | 0.398 | 0.678 | 0.645 | 0.786 |
| | Bootstrapping [22] | 0.397 | 0.677 | 0.640 | 0.784 |
| | HMR [17] | 0.378 | 0.607 | 0.472 | 0.697 |
| | SPIN [20] | 0.420 | 0.591 | 0.391 | 0.656 |
| | Ours | **0.432** | **0.693** | **0.653** | **0.790** |

Table 2: We performance cross-method evaluation on 3DPW dataset and report performance on dense keypoint accuracy.

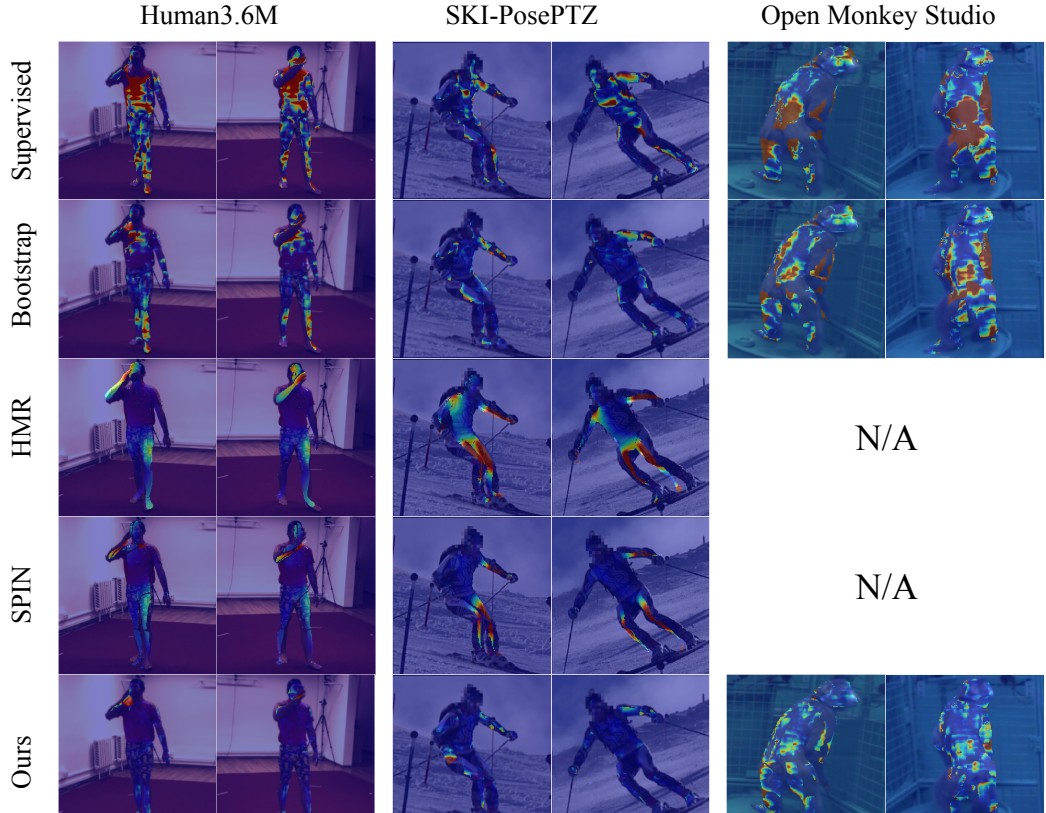

Figure 5: Qualitative results on Human3.6M, Ski-Pose PTZ-Camera and OpenMonkeyPose Datasets. Heatmaps overlapping on images indicate epipolar error for each pixels.

rics, i.e., training on DensePose-COCO
with Human3.6M multiview supervision
and testing on 3DPW without an adapta-
tion, as shown in Table 2. The results show that our trained model with geometric consistency is more generalizable than the baselines.

### 4.7 Evalution on Ski-Pose PTZ-Camera Dataset and OpenMonkeyPose Dataset

We evaluate our method on multiview in-the-wild datasets: Ski-pose and OpenMonkeyPose. Since no ground truth is available, we evaluate geometric consistency summarized in Table 3 and show qualitative results (2nd and 3rd columns in Figure 5). The results show that our model outperforms other baselines by a large margin: 37.2%-54.1% on Ski-Pose and 31.9%-47.5% on OpenMonkeyPose, which can be visually identified by qualitative results.

| Method | Ski-Pose [40] | Monkey [3] |
|---|---|---|
| Supervised | 11.38 | 11.83 |
| Bootstrap [22] | 8.64 | 9.12 |
| HMR [17] | 13.40 | N/A |
| SPIN [20] | 11.83 | N/A |
| Our | **5.43** | **6.21** |

Table 3: Comparison on geometry consistency for in-the-wild data (Epipolar error unit: pixel).

## 5   Conclusion

We present a novel end-to-end semi-supervised approach to learn a dense keypoint detector by leverage a large amount of unlabeled multiview images. Due to the nature of continuous keypoint representation, finding exact correspondences between views is challenging unlike sparse keypoints. We address this challenge by formulating a new dense epipolar constraint that allows measuring a field-to-field geometric error without knowing exact correspondences. Additionally, we proposed a distillation-based regularization to prevent degenerated cases. We design a new network architecture made of twin networks that can effectively measure the dense epipolar error by considering all possible correspondences using affinity matrices. We show that our method outperforms the baseline approaches in keypoint accuracy, multiview consistency, and reconstruction accuracy.

## Acknowledgement

This project is partially supported by NSF IIS 1846031 and NSF IIS 2022894.

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
