# Supplementary Material
# Semi-supervised Dense Keypoints
# using Unlabeled Multiview Images

**Zhixuan Yu**
University of Minnesota
yu000064@umn.edu

**Haozheng Yu**
University of Minnesota
yu000424@umn.edu

**Long Sha**
TuSimple
long.sha@tusimple.ai

**Sujoy Ganguly**
Unity
sujoy.ganguly@unity3d.com

**Hyun Soo Park**
University of Minnesota
hspark@umn.edu

## A   More Implementation Details

Image data is pre-processed in the standard way: cropping, resizing, and then normalizing using the mean and standard deviation of RGB values of ImageNet dataset. We did not apply any data augmentations. For the Human3.6M dataset, we use data from all activities in 1Hz and select camera (0, 2) and (1, 3) to form view pairs. For the Ski-Pose PTZ-Camera dataset, we use 6 adjacent view pairs, i.e. (0, 1), (1, 2), (2, 3), (3, 4), (4, 5) and (5, 1). For the OpenMonkeyPose dataset, we use 4 adjacent view pairs as well.

In the first training stage, we only use $\mathcal{L}_{\mathrm{L}}$. In the second training stage, we use the following loss weights: $\lambda_{\mathrm{L}} = 0$, $\lambda_{\mathrm{M}} = 1$, $\lambda_{\mathrm{R}} = 2000$, $\lambda_{\mathrm{T}} = 10$. We use mini-batches of size 16, each one containing a pair of images. We train our model for 5 epochs using Adam optimizer [1] with a learning rate of $10^{-4}$ on a single NVIDIA Tesla V100-SXM2 GPU with 32.5G memory.

## B   More Ablation Study

We set $\sigma = 1.318e^{-2}$ in equation (4) and $\epsilon = 0.03$ equation (8). We perform ablation study for those two parameters and report results in Table 1. Note that for this experiment we only using "Walking" activity of Human3.6M dataset.

| Method | Keypoint accuracy | | | | Geom. consistency | Recon. accruacy | |
|---|---|---|---|---|---|---|---|
| | AUC$_{10}$ | AUC$_{30}$ | mRCI | mGPS | Epi. error | MPVPE | mMVS |
| $\epsilon = 0.02$ | 0.518 | 0.762 | 0.783 | 0.868 | 2.14 | 51.38 | 0.593 |
| $\epsilon = 0.03$ | 0.530 | 0.766 | 0.783 | 0.869 | 2.08 | 50.74 | 0.600 |
| $\epsilon = 0.05$ | 0.519 | 0.762 | 0.783 | 0.868 | 2.12 | 51.87 | 0.603 |
| $\sigma \to \infty$ | 0.483 | 0.749 | 0.777 | 0.864 | 5.34 | 58.97 | 0.519 |
| $\sigma = 2.402e^{-2}$ | 0.509 | 0.759 | 0.783 | 0.867 | 2.12 | 52.64 | 0.601 |
| $\sigma = 1.908e^{-2}$ | 0.525 | 0.764 | 0.784 | 0.868 | 2.13 | 51.14 | 0.600 |
| $\sigma = 1.576e^{-2}$ | 0.522 | 0.763 | 0.783 | 0.868 | 2.11 | 50.46 | 0.603 |
| $\sigma = 1.318e^{-2}$ | 0.530 | 0.766 | 0.783 | 0.869 | 2.08 | 50.74 | 0.600 |
| $\sigma = 1.084e^{-2}$ | 0.527 | 0.765 | 0.785 | 0.869 | 2.03 | 50.39 | 0.595 |
| $\sigma = 9.320e^{-3}$ | 0.527 | 0.765 | 0.784 | 0.868 | 2.05 | 50.97 | 0.593 |
| $\sigma = 7.610e^{-3}$ | 0.532 | 0.766 | 0.784 | 0.868 | 2.11 | 50.21 | 0.589 |

Table 1: Additionaly ablation study for $\sigma$ and $\epsilon$ on Human3.6M dataset and report performance on keypoint accuracy, geometric consistency and reconstruction accuracy. (Epipolar error unit: pixel; MPVPE unit: mm)

35th Conference on Neural Information Processing Systems (NeurIPS 2021), Sydney, Australia.

| Human3.6M | SKI-PosePTZ | Open Monkey Studio |

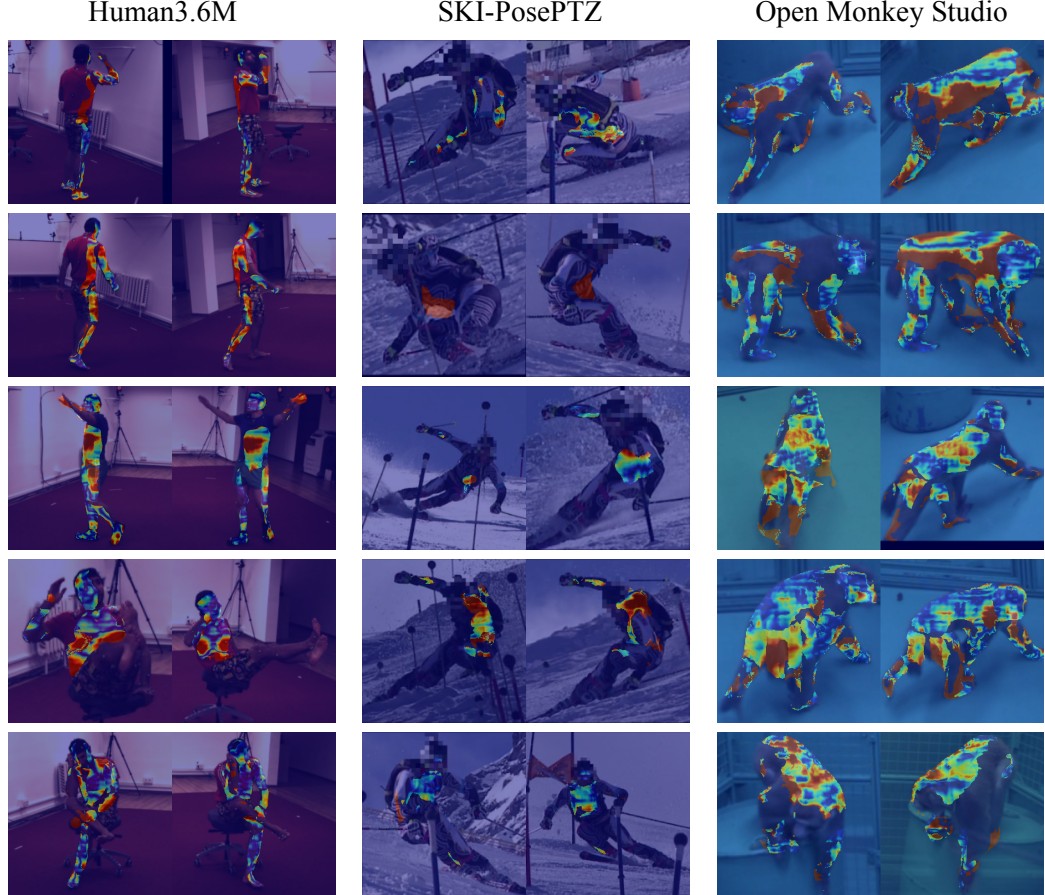

Figure 1: Failure cases on Human3.6M, Ski-Pose PTZ-Camera and OpenMonkeyPose Datasets. Heatmaps overlapping on images indicate epipolar error for each pixels.

## C  Limitations

We characterize some failure cases in terms of geometric consistency, as shown in Figure 1. Our approach fails when the following assumptions do not hold. (1) Relative camera pose (i.e., fundamental matrix) must be accurate. In the last two rows of the second column in Figure 1, the camera poses are inaccurate where epipolar constraint provides misguidance to supervise the dense keypoints. (2) There must be enough corresponding points between two views. In the first 3 rows of first 2 columns of Figure 1, two cameras capture very different views of the target, resulting in very small area visible from both views. (3) The initial dense keypoint field must be reasonably accurate. In the last column of Figure 1, initial dense keypoint field are not accurate enough because poses are from out-of-sample distribution. Besides, samples in the last 2 rows of the first column of Figure 1 correspond to poses do not happen very often in the training set.

## D  Discussion

In this section, we briefly discuss two aspects that we can improve our method.

**Leveraging continuity of correspondence** Consider a pair of corresponding epipolar lines on two images, correspondences on those two lines are expected to be continuous. Currently our method does not enforce this continuity constraint, but it can potentially be used to form an additional supervision signal for dense keypoint detection.

**Use other color space** Currently our multiview photometric consistency loss assumes ambient light and thus enforce RGB values of corresponding points to be the same. This is fine for datasets

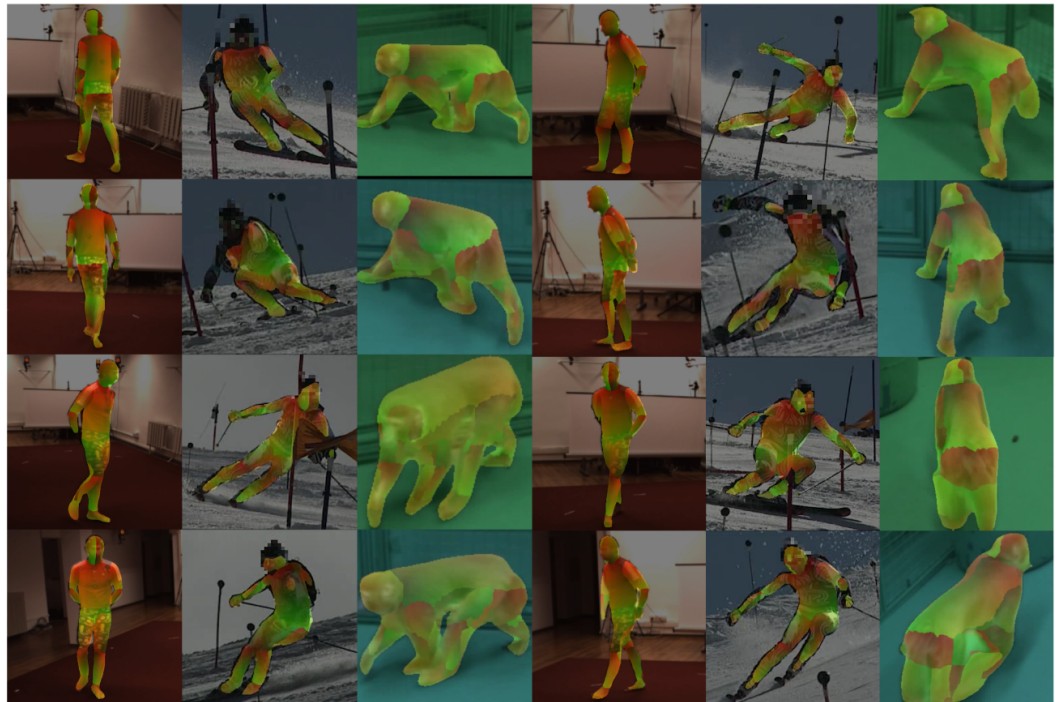

Figure 2: Dense keypoint detection results. Predicted canonical body surface coordinates are color coded by mapping to green and red channels.

without dramatic lighting changes between views, e.g. Human3.6M, but may be too strict in general. Instead, enforcing photometric consistency between correspondences for color components other than lightness makes more sense. This can be achieved by other color space with lightness in a separate channel, e.g. HSV or LAB.

# E    More Results

Here we show more dense keypoint detection results in Figure 2.

# References

[1]  D. P. Kingma and J. Ba. Adam: A method for stochastic optimization. *arXiv*, 2014.