# OpenReview forum: "Dense Keypoints via Multiview Supervision"
_NeurIPS.cc/2021/Conference — NeurIPS 2021 Spotlight_

### Official Review · Reviewer_FiQR · 2021-07-07

**Rating:** 7
**Confidence:** 4

**Summary:**

The paper proposes a semi-supervised approach for dense keypoints detection (assining a unique ID or a UV coordinate to an image of a human body). The key idea of the paper is an epipolar inconsistency loss function which can be computed and trained with multiview datasets. The approach has been evalauted on multi-view human datasets compared against single-view and multi-view techniques. Experimental results are OK, but not very great.

**Limitations And Societal Impact:**

The paper says that the negaive impact is minimum in onen sentence. It rather focuses on descriving postive aspects. This is not a problem to me at all. I do not see issues on negative impacts from this paper.

**Main Review:**

I was not very excited with the paper. I initially liked 2 ideas of the paper: 1) Using epipolar errors does not require extra labels. 2) The keypoint detection engine uses only a single image, and the model can be used for monocular sequences.

However, I was not really convinced (before seeing results) that the first idea is a good one or not. The epipolar error seems very weak and I was in doubt if this really does anything on top of exact supervision. Since the paper is all about semi-supervised, this additional epipolar loss must add significant performance boost against a system without this epipolar loss. Then, I look at the experimantal results and found that the effects of the epipolar loss is not really verified as I suspected. The ablation study must show their system just without L_M in Table 1. This ablation study basically says that L_M does not matter.

I do not really care about geometric consistency metric (geom consistency metric in table 1 and the entire Table 2), because this is the error which the proposed method seeks to minimize. The real performance should be mesure by the real pin-point answers. On this point, Table 1 (Comparison block) is not very exciting. Ignoring (geom consistency column), the proposed approach is the best in 3 out of 6 columns only. And even where they win, the performance gap is very minor at the point that I would rather suspect the engineering difference (likely authors spent more time tweaking their system on getting good numbers). Overall, the performance improvement does not seem to justify a bit complex technical formulation from not-so-convincing idea.

Lastly, on my second point above, the architecture does not mix 2 images which means that the system can be used for monocular sequences. I thought that this is the exciting piece and I wished to see evaluations on monocular sequences say optical flow datasets. The evaluations are solely based on mult-view datasets, which is understandable as generalization is often a problem and they have to train and test on the same dataset. But, a potentially interesting aspect of the paper diminished and I got less excited.

Overall, I did not find this paper exciting. I started with a strong doubt that the main idea may not be very effective. And a bit complicated thing must happen to implement this idea, but overall, the metrics of the idea are not demonstrated and I beleive that there are no such benefits.

--------------
The rebuttal provided new experimental validations and clarifications. My concenrs are resolved and I support the acceptance of the paper.

**Time Spent Reviewing:**

3

---

> ### Author Response · Authors · 2021-08-10
> **We address the reviewer's concerns on effectiveness of multiview consistency loss (L_M), overall performance and generalizability toward unseen single-view dataset.**
>
> We are grateful for you providing insightful comments and potential improvement of the manuscript.
>
>  - We argue that the multiview consistency ($L_M$) plays a key role. Due to the geometric ambiguity of a solution (L187-188), it alone, however, produces a degenerate solution, requiring a regularization ($L_R$) that constrains the space of the solution near the initial model $\phi_0$. This is analogous to solving a homogeneous system $A\mathbf{x}=0$ by limiting the solution space to $\||\mathbf{x}\||=1$ (without it, there exists a degenerate solution $\mathbf{x}=0$). Similar to the constraint in the homogeneous system, the regularization alone ($L_R$) does not add any value: it is identical to the supervised model in theory, i.e., $L_L = L_L+L_R$. To empirically prove this claim, we conduct an additional ablation study by adding $L_L+L_R$ as shown in Table below. It can be seen that $L_L + L_R$ performs similarly to $L_L$ while $L_L + L_M + L_R$ outperforms $L_L + L_R$ and $L_L$ on all metrics (by a margin up to 6.9% for keypoint accuracy metrics and 9.7% for reconstruction accuracy metrics). Similarly, $L_L + L_M + L_R + L_T$ consistently outperforms $L_L + L_R + L_T$ on all metrics (by a margin up to 5.2% for keypoint accuracy metrics and 4.7% for reconstruction accuracy metrics), which further shows the effects of the epipolar loss.
>
> |     |                                                                                                     |           |           |           |           |            |           |           |
> |:----|:----------------------------------------------------------------------------------------------------|:----------|:----------|:----------|:---------:|:-----------|:----------|:----------|
> |     | Method                                                                                              | AUC\_10   | AUC\_30   | mRCI      |   mGPS    | Epi. error | MPVPE     | mMVS      |
> |     | Supervised ($\\mathcal{L}\_{\\rm L}$)                                                               | 0.48      | 0.748     | 0.777     |   0.864   | 5.54       | 58.71     | 0.521     |
> |     | $\\mathcal{L}\_{\\rm L} + \\mathcal{L}\_{\\rm R}$                                                   | 0.482     | 0.749     | 0.779     |   0.864   | 5.36       | 58.47     | 0.513     |
> |     | $\\mathcal{L}\_{\\rm L} + \\mathcal{L}\_{\\rm M}$                                                   | 0.12      | 0.448     | 0.487     |   0.719   | 1.46       | 176.9     | 0.285     |
> |     | $\\mathcal{L}\_{\\rm L} + \\mathcal{L}\_{\\rm M} + \\mathcal{L}\_{\\rm R}$                          | 0.513     | 0.76      | 0.782     |   0.867   | 2.18       | 55.16     | 0.563     |
> |     | $\\mathcal{L}\_{\\rm L} + \\mathcal{L}\_{\\rm R} + \\mathcal{L}\_{\\rm T}$                          | 0.499     | 0.755     | 0.779     |   0.866   | 4.65       | 52.97     | 0.570     |
> |     | $\\mathcal{L}\_{\\rm L} + \\mathcal{L}\_{\\rm M} + \\mathcal{L}\_{\\rm R} + \\mathcal{L}\_{\\rm T}$ | **0.525** | **0.764** | **0.783** | **0.868** | **2.13**   | **51.17** | **0.597** |
>
>  - We would like to note that in Table 1 (also as shown below), HMR and SPIN are trained **with the 3D ground truth** (pre-estimated mesh) of Human3.6M (L253-256) while other algorithms including ours predict dense keypoints **without the 3D ground truth**. Having the 3D ground truth is a significant advantage while it requires a substantial additional amount of 3D annotation effort. It is expected to outperform the ones without it, in particular, on reconstruction accuracy metrics (as mentioned in L292-294). We included these baselines to provide the upper bound of our semi-supervised method as a reference. Note that despite the lack of the 3D ground truth, ours still outperforms HMR and SPIN (by a margin of up to 67.7% and 25.3% on keypoint accuracy metrics) on three keypoint accuracy metrics. Compared to “Supervised” and “Bootstrapping” which are close to our setup, ours outperform on all metrics (by a margin of up to 9.3% and 19.1% on keypoint accuracy metrics).
>
> |     |               |           |          |           |           |            |           |           |
> |:----|:--------------|:----------|:---------|:----------|:---------:|:-----------|:----------|:----------|
> |     | Method        | AUC\_10   | AUC\_30  | mRCI      |   mGPS    | Epi. error | MPVPE     | mMVS      |
> |     | Supervised    | 0.428     | 0.705    | 0.728     |   0.83    | 6.52       | 65.71     | 0.484     |
> |     | Bootstrapping | 0.393     | 0.683    | 0.707     |   0.820   | 5.73       | 64.12     | 0.513     |
> |     | HMR           | **0.495** | 0.68     | 0.589     |   0.76    | 4.0        | 55.9      | **0.809** |
> |     | SPIN          | 0.456     | 0.615    | 0.44      |   0.685   | 3.26       | **54.22** | 0.712     |
> |     | Ours          | 0.468     | **0.72** | **0.738** | **0.834** | **2.7**    | 58.8      | 0.544     |
>
>
>  - We absolutely agree that our method can be evaluated on a single view keypoint detection dataset other than multiview datasets. We conduct a new comparative evaluation on 3DPW dataset on the keypoint accuracy metrics (3D metrics do not apply), i.e., training on DensePose-COCO with Human3.6M multiview supervision and testing on 3DPW without an adaptation. We compare ours with “Supervised”, “Bootstrapping”, “HMR”, and “SPIN” as shown in Table below. The results show that our trained model with geometric consistency is more generalizable than the baselines.
>
> |     |               |           |           |           |           |     |     |     |     |
> |:----|:--------------|:----------|:----------|:----------|:----------|:---:|:----|:----|:----|
> |     | Method        | AUC\_10   | AUC\_30   | mRCI      | mGPS      |     |     |     |     |
> |     | Supervised    | 0.381     | 0.664     | 0.622     | 0.774     |     |     |     |     |
> |     | Bootstrapping | 0.342     | 0.647     | 0.607     | 0.767     |     |     |     |     |
> |     | HMR           | 0.356     | 0.589     | 0.449     | 0.682     |     |     |     |     |
> |     | SPIN          | 0.402     | 0.575     | 0.375     | 0.644     |     |     |     |     |
> |     | Ours          | **0.413** | **0.678** | **0.632** | **0.778** |     |     |     |     |

---

> > ### Comment · Reviewer_FiQR · 2021-08-18
> > **Thank for extra experimantal validations.**
> >
> > I thank authors for providing extra experimental validations and also clarifications. I now agree with the authors that the idea of the paper is very effective and I support the acceptance of the paper.

---

### Official Review · Reviewer_Ao3W · 2021-07-08

**Rating:** 7
**Confidence:** 4

**Summary:**

The authors propose a technique that exploits multi-view consistency for learning dense correspondences in images coupled with a densepose-like mapping in texture space. They performed ablations and SOTA comparisons on H3.6, and further SOTA comparisons on Ski-pose and Monkey.

**Limitations And Societal Impact:**

I am not sure people tracking cannot be seen has having some impact on society (monitoring, recognition, tracking), and not acknowledging this seems a bit naive.

Limitations are in the supplementary, but a short discussion should be moved to the main paper.

**Main Review:**

The paper proposes an interesting technique that introduces multi-view consistency for dense pose estimation. What I would have liked to see is that multi-view consistency allows one to further increase the precision of computed correspondences, but what the authors discover is that by accounting for multi-view consistency... you get more consistent mappings (duh?) – in other words it is a paper that does well on a metric, because they introduced a metric for which other techniques would fail.
So overall, the emerging question is `why should we care?` and `can you show an application scenario where multi-view consistent keypoints are useful?`.
Other than this core questions, I have other concerns about the paper itemized in the remarks below (to which I expect an answer). Note that I am *more than happy* to raise the score if my concerns are properly addressed (the opposite is also true).

## Core remarks

1. My core complaint relates to eq(5,7), and unless properly addressed would prevent me from letting this paper being accepted. First of all, I was left wondering why you needed both of them until I read L217-L224 (this should be addressed). But most importantly, I do not buy the argument provided by (6), and the massive drop in performance in the ablation in Table.1 (from 0.5 to 0.12).... even more so given that the supervised method performs much better (at 0.48)!!! In particular, (5) provides pointwise supervision to the value of the maps, leaving no affine nullspace in the supervision of $\varphi$. To prove this, let us assume the ideal mapping function is an identity function $\varphi(.)=I(.)$, then your statement would say $I(x)==2I(x)$ which is only true for $x=0$ (hence generally false)? So, while I think this argument is flawed, I can guess what happened in Table.1: the aggregated magnitude of (5) is significantly higher in value than the one in (7) (e.g. you forgot normalization?), and hence you are simply retrieving different pareto-optimal solutions (i.e. the magnitude of $\mathcal{L}_R$ and $\mathcal{L}_M$ are similar, while $\mathcal{L}_L$ is much smaller... in other words $\mathcal{L}_M$ is overpowering $\mathcal{L}_L$, which is also consistent with the fact that the geometric consistency score is the lowest across the entire set of experiments).
1. I find it quite disturbing that the paper continuously refers to "geodesic" distances and metric. More specifically, despite the fact that given a piecewise parameterization of geometry, distances between texture map coordinates $u$ *do not* represent geodesic distances. Even if you used small patches and quasi-isometric parameterization, this would only be a decent approximation in small neighborhoods (as isometric maps are only possible for developable surfaces with zero Gaussian curvature). The fact that [8] was sloppy in employing this term does not justify the entire field to adopt it too.
1. L144: `argmin is not differentiable` this is only partially true, as otherwise any other paper that uses Chamfer Distance would not exist in geometric deep learning? Further, the *necessity* to convert from min to softmin has not been ablated at all... why?
1. L158: It is not clear at all to me where the visibility $v$ comes from? Usually pairwise visibility is ensured by cycle consistency... but I see no such loss. Are you assuming visibility to be provided as part of the dataset for every single image? This is a severe limitation, and should be stated as such.
1. Considering part of the paper is related to uncertainty for keypoints, I would consider appropriate an acknowledgement of works in this area (e.g. UGGLI amongst many others... Google it?).

## Further remarks
1. Sec.4: The writing in the technical section could be improved a lot... it would have been a lot faster to digest eq(2) directly, and saying the first term measures for the difference in how two points mapped in texture space, and the latter measures epipolar correctness, and then diving into details.
1. Table.1: it would have been a lot more informative if the ablation would disable one loss at a time, rather than progressively adding them? Further, Why the scores in Ours(comparison) and Ours(ablation/last row) *do not match*?
1. Fig.1 in print you can barely see anything, recommend increasing the image transparency, or increase brightness
1. L57, L??: you state that you generalize the Samson distance (twice), yet never point out the difference
1. L138: missing eq# and $\Theta$ is undefined
1. L149: what's the `range of the body surface?`
1. L162-166: this seems repeated text with zero new information? (repeated earlier in the main body, as well as in Fig.2)
1. L181, L192: I am not exactly sure what `when DensePose is used the field is defined for each body part` and `we augment the loss for body parts` mean. Are you intending the the function $\Phi$ is actually a collection of mapping functions $\{\Phi_k\}$, one per body part (indexed by `k`)?
1. L203: if that's the cardinality of the foreground pixels, wouldn't that make your method choke in terms of memory on reasonably (e.g. 1MP) size images with $\approx O(1e9^2)$ complexity?
1. L207: `design a new operation` not sure I see anything new, isn't this just a vectorization of the same operation? Seems like an implementation detail more than anything else. Am I mistaken?
1. L207-8: missing equation number!!!
1. L187: infinitely → infinite
1. Tab.1: accruacy → accuracy
1. L173-5: if you use $\underbrace{\mathcal{L}}_{\text{photometric loss}}$ you can omit this piece of text and save some space for more useful insight
1. L131: consider explaining why this mapping is injective in your setting?
1. L289: We evaluation → We evaluate

# UPDATE raised my score 6→7 thanks to the rebuttal+discussion

**Time Spent Reviewing:**

4

---

> ### Author Response · Authors · 2021-08-10
> **We address reviewer's question on the goal, application, and method of our paper.**
>
> We are grateful for providing insightful comments and potential improvement of the manuscript.
> The core contribution of our method is to improve the accuracy of dense keypoint detection by leveraging unlabeled multiview images. We agree that 3D reconstruction and multiview consistency are a byproduct of the method. To address this concern, in addition to detection accuracy on multiview image data, we conduct a new comparative evaluation on 3DPW dataset (single view dense keypoint detection) on the keypoint accuracy metrics (3D metrics do not apply), i.e., training on DensePose-COCO with Human3.6M multiview supervision and testing on 3DPW without an adaptation. We compare ours with “Supervised”, “Bootstrapping”, “HMR”, and “SPIN” as shown in Table below. The results show that our trained model with geometric consistency is more generalizable than the baselines.
>
> |     |               |           |           |           |           |     |     |     |     |
> |:----|:--------------|:----------|:----------|:----------|:----------|:---:|:----|:----|:----|
> |     | Method        | AUC\_10   | AUC\_30   | mRCI      | mGPS      |     |     |     |     |
> |     | Supervised    | 0.381     | 0.664     | 0.622     | 0.774     |     |     |     |     |
> |     | Bootstrapping | 0.342     | 0.647     | 0.607     | 0.767     |     |     |     |     |
> |     | HMR           | 0.356     | 0.589     | 0.449     | 0.682     |     |     |     |     |
> |     | SPIN          | 0.402     | 0.575     | 0.375     | 0.644     |     |     |     |     |
> |     | Ours          | **0.413** | **0.678** | **0.632** | **0.778** |     |     |     |     |
>
>
> 1. Equation (6) indicates two facts:
>
>  - There exists a family of solutions $\{\phi\}$ that produces the same loss $L_M$. Consider a distance preserving transformation (e.g., rigid transformation), $T$. There exists two mappings ($\phi, \phi_1$) that result in the same multiview loss, $L_M$: $\||\phi(\mathbf{x}; \mathcal{I}) - \phi(\mathbf{y}; \mathcal{I}')\|| = \||T\phi(\mathbf{x}; \mathcal{I}) - T\phi(\mathbf{y}; \mathcal{I}')\|| = \||\phi_1(\mathbf{x}; \mathcal{I}) - \phi_1(\mathbf{y}; \mathcal{I}')\||$ where $\mathbf{x}$ and $\mathbf{y}$ are a correspondence in image $\mathcal{I}$ and $\mathcal{I}'$, and $\phi_1 = T\phi$. Another example is that $\phi$ is identity, one can learn another $\phi_1$ that produces the same $L_M$, e.g., $\phi_1 = \phi + 2$. Note that there is a typo: it must be a distance preserving transformation not affine transformation.
>
>  - There exists a degenerate solution such as $\phi(\mathbf{x}; \mathcal{I}) = \infty$. When $\phi(\mathbf{x}; \mathcal{I}) = \infty$, it is not visible from the other image ($v(\mathbf{x},I’)=0$), and therefore, $L_M \rightarrow 0$.
> Due to this ambiguity of a solution, $L_M$ alone can produce a degenerate solution, requiring a regularization ($L_R$) that constrains the space of the solution near the initial model $\phi_0$. This is analogous to solving a homogeneous system $A\mathbf{x}=0$ by limiting the solution space to $\||\mathbf{x}\||=1$ (without it, there exists a degenerate solution $\mathbf{x}=0$). Similar to the constraint in the homogeneous system, the regularization alone ($L_R$) does not add any value: it is identical to the supervised model in theory, i.e., $L_L = L_L+L_R$. To empirically prove this claim, we conduct an additional ablation study by adding $L_L+L_R$ as shown in below Table. It can be seen that $L_L + L_R$ performs similarly to $L_L$ while $L_L + L_M + L_R$ outperforms $L_L + L_R$ and $L_L$ on all metrics (by a margin up to 6.9% for keypoint accuracy metrics and 9.7% for reconstruction accuracy metrics). Similarly, $L_L + L_M + L_R + L_T$ consistently outperforms $L_L + L_R + L_T$ on all metrics (by a margin up to 5.2% for keypoint accuracy metrics and 4.7% for reconstruction accuracy metrics), which further shows the effects of the epipolar loss.
>
> 2. We agree. We will not use the term *geodesic*.
>
> 3. The argmin operation is strictly non-differentiable. We address this by using matchability. The soft-argmin can relax the non-differentiability while it is known to be sensitive to spurious detection. We conduct a new evaluation to compare with soft-argmin in Table below. It can be seen that there is a slight performance drop for some metrics.
>
> 4. We acknowledge that the description of the visibility deserves clarity. We will add the following clarification:
> “The visibility is computed on-the-fly using predicted dense keypoints $\phi$. It is visible if there exists a corresponding $\mathbf{u}’ = \phi(\mathbf{y}; \mathcal{I}’)$ from view $\mathcal{I}’$ to the predicted $\mathbf{u} = \phi(\mathbf{x}; \mathcal{I})$ from image I, i.e., $\exists \mathbf{u}' \in \Omega, \|\mathbf{u}’ - \mathbf{u}\| < \epsilon$ where $\epsilon$ is correspondence tolerance defined in the canonical UV space." We use $\epsilon=0.03$ in the paper. We conduct an additional study on two different tolderances 0.02 and 0.05 and report results in the following table. It can be seen that the performance for those 3 different thresholds are similar, but models trained with threshold 0.03 outperform the other two on AUC_10 metric by a margin of 1.3%.
>
> 5. We will add the following references related to uncertainty for keypoints:
>
>  - Kumar, Abhinav, et al. "UGLLI face alignment: Estimating uncertainty with gaussian log-likelihood loss." Proceedings of the IEEE/CVF International Conference on Computer Vision Workshops. 2019.
>
>  - Kumar, Abhinav, et al. "LUVLi Face Alignment: Estimating Landmarks' Location, Uncertainty, and Visibility Likelihood." Proceedings of the IEEE/CVF Conference on Computer Vision and Pattern Recognition. 2020.
>
>
> |     |                                                                                                                        |           |           |           |           |            |           |           |
> |:----|:-----------------------------------------------------------------------------------------------------------------------|:----------|:----------|:----------|:---------:|:-----------|:----------|:----------|
> |     | Method                                                                                                                 | AUC\_10   | AUC\_30   | mRCI      |   mGPS    | Epi. error | MPVPE     | mMVS      |
> |     | Supervised ($\\mathcal{L}\_{\\rm L}$)                                                                                  | 0.48      | 0.748     | 0.777     |   0.864   | 5.54       | 58.71     | 0.521     |
> |     | $\\mathcal{L}\_{\\rm L} + \\mathcal{L}\_{\\rm R}$                                                                      | 0.482     | 0.749     | 0.779     |   0.864   | 5.36       | 58.47     | 0.513     |
> |     | $\\mathcal{L}\_{\\rm L} + \\mathcal{L}\_{\\rm M}$                                                                      | 0.12      | 0.448     | 0.487     |   0.719   | 1.46       | 176.9     | 0.285     |
> |     | $\\mathcal{L}\_{\\rm L} + \\mathcal{L}\_{\\rm M} + \\mathcal{L}\_{\\rm R}$                                             | 0.513     | 0.76      | 0.782     |   0.867   | 2.18       | 55.16     | 0.563     |
> |     | $\\mathcal{L}\_{\\rm L} + \\mathcal{L}\_{\\rm M} + \\mathcal{L}\_{\\rm R} (\\rm{soft-argmax})$                         | 0.505     | 0.757     | 0.782     |   0.867   | 2.22       | 55.76     | 0.548     |
> |     | $\\mathcal{L}\_{\\rm L} + \\mathcal{L}\_{\\rm R} + \\mathcal{L}\_{\\rm T}$                                             | 0.499     | 0.755     | 0.779     |   0.866   | 4.65       | 52.97     | 0.570     |
> |     | $\\mathcal{L}\_{\\rm L} + \\mathcal{L}\_{\\rm M} + \\mathcal{L}\_{\\rm R} + \\mathcal{L}\_{\\rm T}$                    | **0.525** | **0.764** | **0.783** | **0.868** | **2.13**   | **51.17** | **0.597** |
> |     | $\\mathcal{L}\_{\\rm L} + \\mathcal{L}\_{\\rm M} + \\mathcal{L}\_{\\rm R} + \\mathcal{L}\_{\\rm T} (\\epsilon = 0.02)$ | 0.518     | 0.762     | 0.783     |   0.868   | 2.14       | 51.38     | 0.593     |
> |     | $\\mathcal{L}\_{\\rm L} + \\mathcal{L}\_{\\rm M} + \\mathcal{L}\_{\\rm R} + \\mathcal{L}\_{\\rm T} (\\epsilon = 0.05)$ | 0.519     | 0.762     | 0.783     |   0.868   | 2.12       | 51.87     | 0.603     |
>
> Further Remarks:
>
> 1, 3, 5, 7, 9, 10, 11, 12, 13, 14, 15, 16: We will correct typographic errors and add clarification on the suggested comments.
>
> 2. We conduct a new experiment for ablation analysis to study the impact of each loss as shown in Table above (addressed in point 1 above as well). To study each modular term, we leverage a smaller set of Human3.6M data. We will add specifications of the dataset used for ablation study.
>
> 4. The following statement will be added: ”Sampson distance is defined between explicit point correspondence. We generalize it to model probabilistic correspondence.”
>
> 6. The pixel range that corresponds to the body surface (valid range of UV coordinate). It is equivalent to body segmentation.
>
> 8. In practice, we use per part UV map, and therefore, $\phi$ is made of a collection of $\phi_i$ where i is the body part index.

---

> > ### Comment · Reviewer_Ao3W · 2021-08-11
> > **Replies^2**
> >
> > 1) subpoints
> >   - now I understand what you meant, but this was not clear at all... especially as once you introduce eq5, the nullspace is already "gone"; this needs a careful rewrite
> >   - that's exactly my point, just as you say `the regularization alone does not add any value: it is identical to the supervised model in theory` so (5) and (7) are performing the same task... why? *In the table above I would have wanted to see a line where you have the full model and once you ONLY disable L_r, and once you ONLY disable L_l. Having two losses to regularize the same behavior is just odd. And please note that a bold number when both are active is not sufficient... there must be a clear technical reason?* This still needs to be addressed to "clear" the paper from a  technical correctness standpoint.
> >
> > 3) apologies, this was my error as I was thinking of min, not argmin
> >
> > 4) not including this in the paper was a major oversight
> >
> >
> > Other than this, it seems I share my doubts about the point of having `multi-view consistency` with reviewer **FiQR**...
> > I am curious to see how the discussion evolves on that thread (no need to continue here).

---

> > > ### Author Response · Authors · 2021-08-11
> > > **Clarity**
> > >
> > > As we respond earlier, we will revise the manuscript to clarify the geometric ambiguity and visibility, by including mathematical formal definition and examples.
> > >
> > > Note that Equation (5) is not regularization. It is a supervised loss on **the labeled data** that minimizes the error between the ground truth and prediction. On the other hand, Equation (6) is a regularization on **unlabeled data** based on the pseudo label predicted by the pretrained model (similar to model distillation). Please see the two step training procedure in L217-223. These two are designed for different purposes while acting similarly without multiview loss ($L_R$) because the pretrained model used for distillation is pretrained by minimizing supervised loss. When $L_R$ is introduced, they act differently. $L_L+L_M$ find a solution that may overfit to the labeled data while producing a degenerate prediction for the unlabeled data, depending on the proportion of the labeled and unlabeled data (if the unlabeled data dominate, it tends to converge to the degenerate case.). $L_R$ acts on the unlabeled data, which prevents from the degenerate case regardless of the data proportion.
> > >
> > > This is a semi-supervised learning framework that leverages both labeled and unlabeled data by definition. Therefore, the supervised loss on the labeled data ($L_L$) is always necessary. But as suggested, we will post a new experiment on the full model without regularization shortly: $L_L+L_M+L_T$.
> > >
> > > |     |                                                                                                                        |           |           |           |           |            |           |           |
> > > |:----|:-----------------------------------------------------------------------------------------------------------------------|:----------|:----------|:----------|:---------:|:-----------|:----------|:----------|
> > > |     | Method                                                                                                                 | AUC\_10   | AUC\_30   | mRCI      |   mGPS    | Epi. error | MPVPE     | mMVS      |
> > > |     | Supervised ($\\mathcal{L}\_{\\rm L}$)                                                                                  | 0.48      | 0.748     | 0.777     |   0.864   | 5.54       | 58.71     | 0.521     |
> > > |     | $\\mathcal{L}\_{\\rm L} + \\mathcal{L}\_{\\rm R}$                                                                      | 0.482     | 0.749     | 0.779     |   0.864   | 5.36       | 58.47     | 0.513     |
> > > |     | $\\mathcal{L}\_{\\rm L} + \\mathcal{L}\_{\\rm M}$                                                                      | 0.12      | 0.448     | 0.487     |   0.719   | 1.46       | 176.9     | 0.285     |
> > > |     | $\\mathcal{L}\_{\\rm L} + \\mathcal{L}\_{\\rm M} + \\mathcal{L}\_{\\rm R}$                                             | 0.513     | 0.76      | 0.782     |   0.867   | 2.18       | 55.16     | 0.563     |
> > > |     | $\\mathcal{L}\_{\\rm L} + \\mathcal{L}\_{\\rm M} + \\mathcal{L}\_{\\rm R} (\\rm{soft-argmax})$                         | 0.505     | 0.757     | 0.782     |   0.867   | 2.22       | 55.76     | 0.548     |
> > > |     | $\\mathcal{L}\_{\\rm L} + \\mathcal{L}\_{\\rm R} + \\mathcal{L}\_{\\rm T}$                                             | 0.499     | 0.755     | 0.779     |   0.866   | 4.65       | 52.97     | 0.570     |
> > > |     | $\\mathcal{L}\_{\\rm L} + \\mathcal{L}\_{\\rm M} + \\mathcal{L}\_{\\rm T}$                                             | 0.131     | 0.476     | 0.526     |   0.739   | 1.48       | 138.94    | 0.420     |
> > > |     | $\\mathcal{L}\_{\\rm L} + \\mathcal{L}\_{\\rm M} + \\mathcal{L}\_{\\rm R} + \\mathcal{L}\_{\\rm T}$                    | **0.525** | **0.764** | **0.783** | **0.868** | **2.13**   | **51.17** | **0.597** |
> > > |     | $\\mathcal{L}\_{\\rm L} + \\mathcal{L}\_{\\rm M} + \\mathcal{L}\_{\\rm R} + \\mathcal{L}\_{\\rm T} (\\epsilon = 0.02)$ | 0.518     | 0.762     | 0.783     |   0.868   | 2.14       | 51.38     | 0.593     |
> > > |     | $\\mathcal{L}\_{\\rm L} + \\mathcal{L}\_{\\rm M} + \\mathcal{L}\_{\\rm R} + \\mathcal{L}\_{\\rm T} (\\epsilon = 0.05)$ | 0.519     | 0.762     | 0.783     |   0.868   | 2.12       | 51.87     | 0.603     |

---

> > > > ### Comment · Reviewer_Ao3W · 2021-08-23
> > > > **why are titles mandatory?**
> > > >
> > > > It seems you are trying to correct my understanding, but what you wrote is perfectly clear to me.
> > > > While it is true that they do it in a different way, both $L_R$ and $L_L$ exist with the purpose of avoiding a trivial solution to the problem (one is for labelled, the other for unlabelled data).
> > > >
> > > > In the table above, these are the tree most important rows
> > > > 1) $L_L +L_M +L_R +L_T$ (already there)
> > > > 2) $L_L +L_M +L_T$ (already there)
> > > > 3) $L_M +L_R +L_T$
> > > >
> > > > **the last (still missing) is important because it tells you that you need to have some form of fully supervised data.** As soon as you can confirm this quantitatively, and given the recent reply by FiQR
> > > > about the performance, I'll raise my score.

---

> > > > > ### Author Response · Authors · 2021-08-23
> > > > > **One more ablation study added (L_M + L_R + L_T)**
> > > > >
> > > > > As suggested, we conduct an additional ablation study for $\mathcal{L}_M + \mathcal{L}_R + \mathcal{L}_T$ and report in the following table. As you can see, it completely fails without fully supervised data, which is consistent with what we expected.
> > > > >
> > > > > |     |                                                                                                                        |           |           |           |           |            |           |           |
> > > > > |:----|:-----------------------------------------------------------------------------------------------------------------------|:----------|:----------|:----------|:---------:|:-----------|:----------|:----------|
> > > > > |     | Method                                                                                                                 | AUC\_10   | AUC\_30   | mRCI      |   mGPS    | Epi. error | MPVPE     | mMVS      |
> > > > > |     | Supervised ($\\mathcal{L}\_{\\rm L}$)                                                                                  | 0.48      | 0.748     | 0.777     |   0.864   | 5.54       | 58.71     | 0.521     |
> > > > > |     | $\\mathcal{L}\_{\\rm L} + \\mathcal{L}\_{\\rm R}$                                                                      | 0.482     | 0.749     | 0.779     |   0.864   | 5.36       | 58.47     | 0.513     |
> > > > > |     | $\\mathcal{L}\_{\\rm L} + \\mathcal{L}\_{\\rm M}$                                                                      | 0.12      | 0.448     | 0.487     |   0.719   | 1.46       | 176.9     | 0.285     |
> > > > > |     | $\\mathcal{L}\_{\\rm L} + \\mathcal{L}\_{\\rm M} + \\mathcal{L}\_{\\rm R}$                                             | 0.513     | 0.76      | 0.782     |   0.867   | 2.18       | 55.16     | 0.563     |
> > > > > |     | $\\mathcal{L}\_{\\rm L} + \\mathcal{L}\_{\\rm M} + \\mathcal{L}\_{\\rm R} (\\rm{soft-argmax})$                         | 0.505     | 0.757     | 0.782     |   0.867   | 2.22       | 55.76     | 0.548     |
> > > > > |     | $\\mathcal{L}\_{\\rm L} + \\mathcal{L}\_{\\rm R} + \\mathcal{L}\_{\\rm T}$                                             | 0.499     | 0.755     | 0.779     |   0.866   | 4.65       | 52.97     | 0.570     |
> > > > > |     | $\\mathcal{L}\_{\\rm M} + \\mathcal{L}\_{\\rm R} + \\mathcal{L}\_{\\rm T}$                                             | 0.006     | 0.074     | 0.000     |   0.222   | 5.78       | 455.9     | 0.002     |
> > > > > |     | $\\mathcal{L}\_{\\rm L} + \\mathcal{L}\_{\\rm M} + \\mathcal{L}\_{\\rm T}$                                             | 0.131     | 0.476     | 0.526     |   0.739   | 1.48       | 138.94    | 0.420     |
> > > > > |     | $\\mathcal{L}\_{\\rm L} + \\mathcal{L}\_{\\rm M} + \\mathcal{L}\_{\\rm R} + \\mathcal{L}\_{\\rm T}$                    | **0.525** | **0.764** | **0.783** | **0.868** | **2.13**   | **51.17** | **0.597** |
> > > > > |     | $\\mathcal{L}\_{\\rm L} + \\mathcal{L}\_{\\rm M} + \\mathcal{L}\_{\\rm R} + \\mathcal{L}\_{\\rm T} (\\epsilon = 0.02)$ | 0.518     | 0.762     | 0.783     |   0.868   | 2.14       | 51.38     | 0.593     |
> > > > > |     | $\\mathcal{L}\_{\\rm L} + \\mathcal{L}\_{\\rm M} + \\mathcal{L}\_{\\rm R} + \\mathcal{L}\_{\\rm T} (\\epsilon = 0.05)$ | 0.519     | 0.762     | 0.783     |   0.868   | 2.12       | 51.87     | 0.603     |

---

> > > > > > ### Comment · Reviewer_Ao3W · 2021-08-24
> > > > > > **title title title**
> > > > > >
> > > > > > Excellent, as promised I'll now raise my score to "7:accept".
> > > > > > Would be good to articulate why such a total collapse takes places, I was expecting something a bit less drastic :)

---

### Official Review · Reviewer_xjUf · 2021-07-14

**Rating:** 6
**Confidence:** 3

**Summary:**

This paper proposes a new method for the semi-supervised learning of keypoints using multiview images. The main challenge addressed is finding exact correspondences betweenkeypoints in multiple views, since the inverse of keypoint matching cannot be analytically derived or differentiated. The proposed probabilistic epipolar constraint encodes a soft correspondence and geometric consistency in the correspondence field. A distillation-based regularization is also proposed to avoid degenerative cases. Each contribution is ablated and show significant improvement over the baseline and relative to other methods.

**Limitations And Societal Impact:**

The authors mention scalable behavioral analysis as a potential societal impact of this work, since it will enable the study of animal life's individual and social behaviors by eliminating the need of annotated data for image processing.

**Main Review:**

This paper's contributions are:

- A novel formulation of probabilistic epipolar constraints that can be used to enforce multiview consistency on continuous dense keypoints fields
- A new neural network design that enable the precise measurement of of probabilistic epipolar error
- A distillation-based regularization term
- Strong performance on real-world multiview image data

I believe this paper makes a convincing argument for all these claimed contributions, however some further clarification in some points would be useful for my final decision, as described below:

The idea of probabilistic epipolar constraints is well-motivated in terms of multiview keypoint matching, and is well positioned relative to other methods in the related work section. The motivation behind precise measurement of epipolar error, although not particularly novel, has many practical applications beyond improving performance, as a tool for uncertainty estimation.

The distillation-based regularization term addresses a practical limitation of the method, and multiple papers have already explored distillation / teacher networks to improve the performance of multi-frame networks in such degenerate cases. Although seems like the coefficient for this loss has to be carefully determined, so the network is allowed to deviate from it starting values, but not too much in the degenerate cases. Some analysis on the impact of varying this loss would have been interesting. Is there any sort of masking to determine in which areas this regularization should be applied, or is it applied equally?

Performance is evaluated on multiple large-scale datasets, with several baselines (including [22], the method that more closely relates to the setting the proposed method operates on), metrics, and qualitative results. The ablation table shows the impact of each component, however seems like L_T (photometric consistency loss) has very minor impact on the final performance. The same can be said for the generalization performance and comparison with other methods, improvement seems to be much higher for geometric and reconstruction accuracy than for keypoint accuracy. Some reasoning as to why that's the case would be interesting.



**Time Spent Reviewing:**

2

---

> ### Author Response · Authors · 2021-08-10
> **We address reviewer's questions on coefficient of regularization loss, concerns on performance and usefulness of our method.**
>
> We are grateful for you providing insightful comments and potential improvement of the manuscript.
>
>  - We choose the coefficient for this regularization loss empirically, i.e. start from 0 and gradually increase it until we do not observe the performance gain of the model. We will add a plot showing the performance change as varying the coefficient of this loss in the updated manuscript. We apply this regularization to foreground pixels (background pixels are masked out) equally. We will add a discussion on distillation based approaches and their weight controls.
>
>  - Though additionally using $\mathcal{L}_\rm{T}$ does not help much for keypoint accuracy and geometric consistency (up to ~2%), there is a clear improvement for the reconstruction accuracy (by 6%-7%). We would like to note that in Table 1 (also as shown below), HMR and SPIN are trained **with the 3D ground truth** (pre-estimated mesh) of Human3.6M (L253-256) while other algorithms including ours predict dense keypoints **without the 3D ground truth**. Having the 3D ground truth is a significant advantage while it requires a substantial additional amount of 3D annotation effort. It is expected to outperform the ones without it, in particular, on reconstruction accuracy metrics (as mentioned in L292-294). We included these baselines to provide the upper bound of our semi-supervised method as a reference. Note that despite the lack of the 3D ground truth, ours still outperforms HMR and SPIN (by a margin of up to 67.7% and 25.3% on keypoint accuracy metrics) on three keypoint accuracy metrics. Compared to “Supervised” and “Bootstrapping” which are close to our setup, ours outperform on all metrics (by a margin of up to 9.3% and 19.1% on keypoint accuracy metrics).
>
> |     |               |           |          |           |           |            |           |           |
> |:----|:--------------|:----------|:---------|:----------|:---------:|:-----------|:----------|:----------|
> |     | Method        | AUC\_10   | AUC\_30  | mRCI      |   mGPS    | Epi. error | MPVPE     | mMVS      |
> |     | Supervised    | 0.428     | 0.705    | 0.728     |   0.83    | 6.52       | 65.71     | 0.484     |
> |     | Bootstrapping | 0.393     | 0.683    | 0.707     |   0.820   | 5.73       | 64.12     | 0.513     |
> |     | HMR           | **0.495** | 0.68     | 0.589     |   0.76    | 4.0        | 55.9      | **0.809** |
> |     | SPIN          | 0.456     | 0.615    | 0.44      |   0.685   | 3.26       | **54.22** | 0.712     |
> |     | Ours          | 0.468     | **0.72** | **0.738** | **0.834** | **2.7**    | 58.8      | 0.544     |
>
>  - To further validate the usefulness of our method in terms of keypoint accuracy, we conduct a new comparative evaluation on 3DPW dataset on the keypoint accuracy metrics (3D metrics do not apply), i.e., training on DensePose-COCO with Human3.6M multiview supervision and testing on 3DPW without an adaptation. We compare ours with “Supervised”, “Bootstrapping”, “HMR”, and “SPIN” as shown in Table below. The results show that our trained model with geometric consistency is more generalizable than the baselines.
> |     |               |           |           |           |           |     |     |     |     |
> |:----|:--------------|:----------|:----------|:----------|:----------|:---:|:----|:----|:----|
> |     | Method        | AUC\_10   | AUC\_30   | mRCI      | mGPS      |     |     |     |     |
> |     | Supervised    | 0.381     | 0.664     | 0.622     | 0.774     |     |     |     |     |
> |     | Bootstrapping | 0.342     | 0.647     | 0.607     | 0.767     |     |     |     |     |
> |     | HMR           | 0.356     | 0.589     | 0.449     | 0.682     |     |     |     |     |
> |     | SPIN          | 0.402     | 0.575     | 0.375     | 0.644     |     |     |     |     |
> |     | Ours          | **0.413** | **0.678** | **0.632** | **0.778** |     |     |     |     |

---

### Official Review · Reviewer_fHWo · 2021-07-14

**Rating:** 7
**Confidence:** 4

**Summary:**

The paper presents a method for learning continuous dense keypoint fields with part-supervised/labelled data and part-unlabelled data. The authors derive a probabilistic matchability constraint and define a loss that includes multi-view consistency, photometric consistency and a supervised loss from the labelled data. The method is evaluated quantitatively by comparing to a few other baselines and doing ablation studies on various loss terms and train/test splits. Geometric consistency and reconstruction accuracy are used as metrics to measure the accuracy of the proposed method and a few qualitative results are provided.

**Limitations And Societal Impact:**

The datasets used are cited but the licenses of the datasets are not cited. A reader can look up the licences though.
The authors do not mention any negative societal impact, which is largely true because the proposed method will be used as a building block in larger systems. It is the larger systems or the applications that may have positive/negative societal impact. However, the limitations of the method could have been made clearer in the paper. A more thorough evaluation will naturally provide the evidence required to draw conclusions about the limitations of the method. Combinations of limitations of the building blocks used in a larger systems can come together to create blind spots in larger systems, so each block has a role to play. Being upfront about the limitations of each method definitely helps assess the impact of larger systems.

**Main Review:**

**Main paper**

1. Motivation: It might be good to elaborate a bit on what the applications of the proposed method would look like, in order to motivate the method soundly. Typically, keypoint matching whether sparse/dense and discete or continuous is used as a first step in learning the transformation between a pair of images. While there are several classical methods to solve this problem, there is also a rich body of learnt methods such as Spatial Transformer Network etc, that learn and predict the transformation directly. Could you provide example scenarios where having keypoint matches (i.e. the intermediate output) is necessary?

2. Related work: What is the downside of applying sparse keypoint matching? Alternatively, what motivates the need for dense keypoint matching? There are many learned keypoint detection/matching methods in literature such as 3DFeatNet, R2D2, SKD, USIP. Would the authors be able to provide some clarity on where these methods fall short (and therefore what the USP of the proposed method is)? In summary, it would be helpful to motivate clearly why dense keypoint matching is needed and sparse matching is not sufficient? Sparse matches help recover an accurate transformation matrix, which gives dense mapping between images.

3. Section 3.2 [Loss definition]: Loss equation could be written out more clearly. Why are some equations missing numbering? It's unclear whether $\lambda_M$ and $\lambda_R$ are inside the summation or pre-multipliers to the next term such as $\mathcal{L}_M$ and $\mathcal{L}_R$. Reading the description later in the paper clarifies this, but it would be good to sort out correct spacing or multi-line equation here to help resolve this.

4. Is there a constraint on $\lambda_*$ as weights. Looking at the supplementary information suggests that there is no constraint and these are multipliers of the implicit weight of $\lambda_L = 1$ for supervised loss $\mathcal{L}_L$. Please clarify why a constraint on the weights is not needed and provide some intuition or explanation for the _very_ different values of the relative weights for the various terms in the loss function.

5. Section 3.2 [Loss definition]: I think multi-view consistency loss is not really working because it is naïve. The true match for a keypoint lies anywhere on the epipolar line if only two images are available. If three images and the fundamental matrix $F$ is available, then a point-point constraint can be derived and used. The equations and training details seem to suggest that only two images are being used. Can the authors confirm if two images or three images are used for multi-view consistency loss? The regularizing loss $\mathcal{L}_R$ is formulated as 'not too far from the pretrained keypoint prediction'. Given that this is defined on labeled data with ground truth, it effectively imposes the multiview consistency constraint correctly and hence improves the results (as evidenced in ablation study on loss terms in Table 4). This may explain why results for $\mathcal{L}_L + \mathcal{L}_M$ are consistently much worse than those for $\mathcal{L}_L$ alone and those for $\mathcal{L}_L + \mathcal{L}_M + \mathcal{L}_R$. I suspect, $\mathcal{L}_M$ can just be dropped and $\mathcal{L}_R$ could be used instead. To this end, it would be interesting to see an ablation for $\mathcal{L}_L + \mathcal{L}_R$. Hence or otherwise, it would be good to see a decent justification for the inclusion of $\mathcal{L}_M$ term in it's curent form. See also my comment regarding Supplementary info Section A.

6. Photometric consistency loss: I wonder if results from an HSV or albedo related colour space insteadof RGB are better in any noticeable way? They may not be, if the network has sufficient capacity to learnt the RGB to HSV conversion implicitly. Allowing the colour components to change less than the lightness or brightness components should lead to better robustness for lighting changes. I suspect this is not an experiment worth doing unless dealing with datasets with extreme lighting changes. Since only a handful of examples of failure case are provided, I leave it to the authors to judge whether this is an area they think needs improvement. The goodness of the term $\mathbb{T}(x,x')$ is entirely dependent on the goodness of the norm $||\mathcal{I}(x)-\mathcal{I'}(x')||$ and what representation is chosen for $\mathcal{I}(x)$ in general. I do acknowledge that changing the representation of $\mathcal{I}(x)$ will affect rest of the network too, so perhaps only the distance metric here should be updated instead.

7. Evaluation: The method is trained on Human3.6M and then evaluated on SkiPose and OpenMonkeyStudio datasets. all three datasets have a reasonably long range or static background and a moving object in the foreground. It could be useful to see results on a different kind of dataset to disamiguate whether the proposed approach relies on the foreground/background setting in order to be sure that the background pixels are also contributing to the learning process. Another way to verify this is to plot saliency maps for network weights. If the method does capture the fact that the foreground moves against a static background, then that would be fine, but then the paper should specify what kinds of applications such a network is suited to. Ultimately, such analysis will help answer the question 'how general is the approach?'.

8. Evaluation: Figure 5: While it's good to see examples of dense keypoint matching results, it's unclear how to interpret the general goodness of the algorithm from the results in Figure 5. An alternative way to show the results could be to show the calculated transformation between the images and possibly a comparison against a pair of images where the ground truth transformation is available. If there are any patterns in the errors, they will likely become clear from these mappings. The continuous dense keypoint field could be sampled on a regular grid in the source image to acquire some grid points in order to enable comparison to classical multiview geometric methods.

9. Results [Table 1]: I am confused about why 'Ours' does better on geodesic point similarity despite being worse on AUC10 compared to HMR. 'Ours' does do better on AUC30 compared to HMR, which suggests that 'Ours' has a flatter error profile but that does not correlate with high geodesic point similarity. It dependes somewhat on the value of $\kappa$ used. Would the authors comment on scenarios where AUC30 is preferred over AUC10? It's a little bit counter intuitive that 'Ours' does better on geodesic point similarity (RCI is just anothermanifestation of GPS) and epipolar error, yet does not do well on AUC10. Evaluating something like AUC15 or AUC20 will help uncover if it's just that the crossover point between HMR and 'Ours' is just past 10cm.

10. Line 142-143: "A point that is not visible in a corresponding image does not have to be forced to match" - A no match found is a valid outcome. This claim is misleading, because authors use this outcome themselves later in the paper.

11. Line 144: I find the argument about geometric error being discrete and hence non-differentiable a bit arduous. The expected geometric error (i.e. the version using argmin) is discrete only by choice. It is perfectly possible to define continuous versions of the same quantity as indeed the authors do later in the paper. But of course, if one wants to select a single correspondence, then we end up with a discontinuity. I am not sure what purpose this point is serving. I suggest rewriting this.

**Supplementary information:**

1. Section A [Training details] mentions you chose $\mathcal{L}_M = 1.0$, $\mathcal{L}_R = 2000$, $\mathcal{L}_T = 10$.
    Did you mean the weights $\lambda_M = 1.0, \lambda_R = 2000, \lambda_T = 10$ instead?
    If yes, then perhaps this supports my suggestion for dropping $\mathcal{L}_M$ term completely. $\mathcal{L}_R$ is the term that is actually imposing the multi-view consistency constraint. I think that $\mathcal{L}_T$ is too strict (or restricitve), and that's why you
    (a) have accuracy issues where the photoconsistency is poor and
    (b) only a small amount of correction is allowed in the training (i.e. $\lambda_T << \lambda_R$)
    This also raises another question that the supervised loss $\mathcal{L}_T$ (aka label loss) has a very small contribution to the total loss, and yet, we see from ablation studies that $\mathcal{L}_T$ alone has reasonable performance.

2. Section A [Training details]: For reproducibility, it would be helpful to have a few more training details such as dataset splits, data augmentations if any, pre-processing of the data, cleaning, imputation and view selection for multi-view consistency loss where appropriate.

3. Section B [Failure cases] provides some explanations for failure cases. I find these puzzling. There must be enough corresponding points between the two views. If you output a probabilitistic matching score, then perhaps it is possible to set a threshold for accepting the matches. For various values of the acceptability threshold, you would be able to determine the fraction of 'accepted matches' vs 'total matches' and also determine the point beyond which the separation in the two views is so large that no acceptable matches can be found. Something like that may be a better measure of the quality of matching rather than saying that the network fails due to one of these three reasons. In other words it would be good to evaluate the failure modes of the dense keypoint matching network more systematically. The separation in views can be measured in terms of the angular difference in 3D. Although the proposed method is for continuous dense keypoints, the said continuous field can be sampled at discrete locations based on pixel locations of the source image and all evaluation metrics for sparse/discrete keypoint methods can then apply.

4. Section C [Further results] provides further dense keypoint matching results in Figure 2. How should the figure be interpreted? Are the corresponding pairs arranged vertically or horizontally? Is the colour map same as that in the paper. A caption, even if it repeats text from the paper, would be helpful here.

**Notes**

1. Equation numbering: It would be good to have _all_ equations or math expressions numbered. The equations between Eq(1) and Eq(2) are not numbered? The loss definition in Section 3.2 is also not numbered.

2. Typos: I think in quite a fewplaces where you refer to 'synchronized images', you actually mean 'corresponding images'.

3. Typos: Line 289: 'We evaluation', Line 296: 'margain', Line 299: 'does not seen any', Line 300: 'still get improved'

4. Typo? Line 251: Am I correct in understanding that authors mean to refer to DensePose-COCO as the first model in the comparison rather than initial model.

**Overall assessment**
The paper presents a new method for continuous dense keypoint matching that leverages existing ideas to build a system with improved accuracy while using unlabelled images for training. The motivation for the work could be substantiated better. The method is evaluated quantitatively; improvement over existing methods is a little incremental, but noticeable nevertheless. A thorough analysis of what makes the system work or not work (limitations) and some comments about the statistical significance of the results would be welcome. There are a few fundamental issues in the development of the theory that are as yet not very well supported by the results or the reasoning presented in the paper.

**Time Spent Reviewing:**

6

---

> ### Author Response · Authors · 2021-08-10
> **We address reviewer's questions on motivation, related works, method details, results and supplementary material of our paper.**
>
> We are grateful for you providing insightful comments and potential improvement of the manuscript.
>
> 1. The main goal of the paper is to learn a dense keypoint detector, which has been used in many applications such as 3D mesh reconstruction, texture/style transfer, and geometry learning (L31-32). Previous works have leveraged a large scale annotated dataset, which is challenging because of visual/geometric ambiguity and prohibitively large annotation effort. This paper addresses this annotation challenge by using unlabeled multiview images. As a byproduct, the method can generate 3D reconstruction.
> This work can be thought of as a line of work that aims to learn a geometrically coherent representation using various transformations. A spatial transformer is one of such kinds that learns the transformation by generating/sampling data or by synthesis. A perspective transformer is another line of work that learns the representation by predicting 3D. Our work bypasses 3D reconstruction/prediction: we leverage the epipolar constraint directly to learn 2D representation, which requires learning fewer parameters. We will include this discussion in the Related work section.
>
> 2. The difference between sparse and dense keypoints is analogous to the difference between 3D sparse point cloud and 3D mesh. The mesh defines the full geometry of an object, not only its pose but also the shape. In particular, sparse points in humans and animals are defined by the landmark location (3D) while dense points are defined by surface and 6D transformation. There are some works that leverage dense points as input or supervision for 3D human shape and pose estimation, e.g. DecoMR, DenseRac.
>
> 3. We will revise the manuscript accordingly.
>
> 4. We choose the weights empirically. However, we found that the overall performance is not sensitive to the choice of the weights. We will add a new figure that illustrates the impact of the weights by comparing the performance (dense keypoint accuracy).
>
> 5. We use two images for computing multi-view consistency loss, which is a minimal configuration to enforce epipolar constraint. Using a three view constraint can be complementary to our approach while more combinations need to be validated if dense keypoint detection error is present (similar to RANSAC).
> We argue that the multiview consistency ($L_M$) plays a key role. Due to the geometric ambiguity of a solution (L187-188), it alone, however, produces a degenerate solution, requiring a regularization ($L_R$) that constrains the space of the solution near the initial model $\phi_0$. This is analogous to solving a homogeneous system $A\mathbf{x}=0$ by limiting the solution space to $\||\mathbf{x}\||=1$ (without it, there exists a degenerate solution $\mathbf{x}=0$). Similar to the constraint in the homogeneous system, the regularization alone ($L_R$) does not add any value: it is identical to the supervised model in theory, i.e., $L_L = L_L+L_R$. To empirically prove this claim, we conduct an additional ablation study by adding $L_L+L_R$ as shown in Table below. It can be seen that $L_L + L_R$ performs similarly to $L_L$ while $L_L + L_M + L_R$ outperforms $L_L + L_R$ and $L_L$ on all metrics (by a margin up to 6.9% for keypoint accuracy metrics and 9.7% for reconstruction accuracy metrics). Similarly, $L_L + L_M + L_R + L_T$ consistently outperforms $L_L + L_R + L_T$ on all metrics (by a margin up to 5.2% for keypoint accuracy metrics and 4.7% for reconstruction accuracy metrics), which further shows the effects of the epipolar loss.
>
> |     |                                                                                                     |           |           |           |           |            |           |           |
> |:----|:----------------------------------------------------------------------------------------------------|:----------|:----------|:----------|:---------:|:-----------|:----------|:----------|
> |     | Method                                                                                              | AUC\_10   | AUC\_30   | mRCI      |   mGPS    | Epi. error | MPVPE     | mMVS      |
> |     | Supervised ($\\mathcal{L}\_{\\rm L}$)                                                               | 0.48      | 0.748     | 0.777     |   0.864   | 5.54       | 58.71     | 0.521     |
> |     | $\\mathcal{L}\_{\\rm L} + \\mathcal{L}\_{\\rm R}$                                                   | 0.482     | 0.749     | 0.779     |   0.864   | 5.36       | 58.47     | 0.513     |
> |     | $\\mathcal{L}\_{\\rm L} + \\mathcal{L}\_{\\rm M}$                                                   | 0.12      | 0.448     | 0.487     |   0.719   | 1.46       | 176.9     | 0.285     |
> |     | $\\mathcal{L}\_{\\rm L} + \\mathcal{L}\_{\\rm M} + \\mathcal{L}\_{\\rm R}$                          | 0.513     | 0.76      | 0.782     |   0.867   | 2.18       | 55.16     | 0.563     |
> |     | $\\mathcal{L}\_{\\rm L} + \\mathcal{L}\_{\\rm R} + \\mathcal{L}\_{\\rm T}$                          | 0.499     | 0.755     | 0.779     |   0.866   | 4.65       | 52.97     | 0.570     |
> |     | $\\mathcal{L}\_{\\rm L} + \\mathcal{L}\_{\\rm M} + \\mathcal{L}\_{\\rm R} + \\mathcal{L}\_{\\rm T}$ | **0.525** | **0.764** | **0.783** | **0.868** | **2.13**   | **51.17** | **0.597** |
>
> 6. We agree that using HSV or albedo related colour space can lead to a stronger performance that is robust to illumination change. We will study the choice of color space in the new supplementary result.
>
> 7. We leverage DensePose IUV map where I map provides body part segmentation from an image. We will include this body part (foreground) segmentation map in the new supplementary material.
>
> 8. We will add a figure showing side-by-side comparison of 3D smpl mesh rendered with texture transferred from images by predicted and ground truth dense keypoints. This will help interpreting the model performance in terms of  the general goodness of the algorithm.
>
> 9. We acknowledge that ours does not outperform baseline methods for every metric. This is possible even if  “Our” outperform HMR on instance-wise accuracy metrics (mRCI & mGPS) in scenarios where, for example, HMR is only very good at some samples (=> higher AUC_10) while failing in many others (=> lower AUC_30, mRCI & mGPS).
>
> We would like to note that in Table 1, HMR and SPIN are trained **with the 3D ground truth** (pre-estimated mesh) of Human3.6M (L253-256) while other algorithms including ours predict dense keypoints **without the 3D ground truth**. Having the 3D ground truth is a significant advantage while it requires a substantial additional amount of 3D annotation effort. It is expected to outperform the ones without it, in particular, on reconstruction accuracy metrics (as mentioned in L292-294). We included these baselines to provide the upper bound of our semi-supervised method as a reference. Note that despite the lack of the 3D ground truth, ours still outperforms HMR and SPIN (by a margin of up to 67.7% and 25.3% on keypoint accuracy metrics) on three keypoint accuracy metrics. Compared to “Supervised” and “Bootstrapping” which are close to our setup, ours outperform on all metrics (by a margin of up to 9.3% and 19.1% on keypoint accuracy metrics).
>
> 10. We will clarify the sentence by adding the following example: “For instance, a point with two neighbours on the opposite sides is forced to be matched to one of them even though it has similar distances from them”.
>
> 11. It is nondifferentiable not because of discrete representation of keypoints but because of argmin (nearest neighbor search) which is strictly non-differentiable. In L144, we would like to highlight the challenge of learning dense correspondences in the presence of argmin operation. We will clarify the sentence by providing in-depth discussion of non-differentiability.
>
> Supplementary Material:
> 1. We will correct the typographic errors. The magnitude of loss weights differs because each loss has a different unit scale and batch size. We use large $\lambda_R$ because the scale of the $\mathcal{L}_\rm{R}$ is significantly smaller than the rest of terms. In the final manuscript, we will add a new analysis and plot on the weight impact on the overall performance.
>
> 2. We will additionally add the following information to the updated manuscript:
>  - Image data is pre-processed in the standard way: cropping, resizing, and then normalization using the mean and standard deviation of RGB values of ImageNet dataset. We did not apply any data augmentations.
>  - For the Human3.6M, view pair (0, 2) and (1, 3) are selected for computing multi-view consistency loss.
>  - For the Ski-Pose PTZ-Camera, we select 6 adjacent view pairs, i.e.  (0, 1), (1, 2), (2, 3), (3, 4), (4, 5) and (5, 1)  for computing multi-view consistency loss.
>  - For the OpenMonkeyPose, we count the number of 2d joints that are lying inside the predicted mask for each image. Only the images with a mask covering 12 joints among 13 will be used for training our network or testing. We split about 64K images for training and 12K images for testing. We select 4 close view pairs for computing multi-view consistency loss.
>
> 3. As suggested, we will characterize the matching failure cases as a function of the camera baseline (3D angle between views). Specifically, we will perform pixel-level (downsampling if needed) analysis and plot curves showing coincidence between angular difference in 3D and evaluation results.
>
> 4. Figure 2 in Section 3 illustrates the learned UV coordinates where U coordinate takes green channel and V coordinate takes red channel, following DensePose color scheme. We will add a caption to clarify the visualization and colormap.
>
> We will correct the suggested typographic errors and clarify unclear statements in the final manuscript.

---

> > ### Comment · Reviewer_fHWo · 2021-08-21
> > **Re: Response to review**
> >
> > Thank you for the clarifications and the extra ablation study. While it is clear that the loss $\mathcal{L}_M$ _should_ have a reasonable contribution to the improvement in performance, in practice, it does not seem to do so on it's own. It does, however, help consistently, when combined with $\mathcal{L}_R$. So I think if the wording in the paper was changed to refer to the combination $\mathcal{L}_M$+$\mathcal{L}_R$, it would be more convincing. The motivation is sound and I agree with the authors, but the results support the argument "loss $\mathcal{L}_M$ combined with $\mathcal{L}_R$" improve results much better than they support the argument "loss $\mathcal{L}_M$" is key to improving performance. I see that this point has been consistently raised by other reviewers as well, so it would do with further clarification or repositioning. If this aspect was addressed clearly, I could be persuaded to up my rating to 7 (accept).
> >
> > With regards to the main contribution of the paper and the motivation for dense keypoint matching, the clarifications from the authors are adequate and clear now. I see self-supervised dense keypoint matching method as the central contribution of the paper. Thank you for providing the new results and details about the experimental setting. I believe these additions will make the paper stronger and communicate the impact of the work clearly.

---

> > > ### Author Response · Authors · 2021-08-21
> > > **Re: Response to reviewer**
> > >
> > > Thank you for your suggestion regarding the wording in the paper about the role of $\mathcal{L}_M$. We agree with this and will modify the manuscript accordingly to emphasize that $\mathcal{L}_M$ alone helps consistency but only when combining $\mathcal{L}_M$ and $\mathcal{L}_R$ the performance on keypoint and reconstruction accuracy will be increased.

---

> > > > ### Comment · Reviewer_fHWo · 2021-08-27
> > > > **Re:**
> > > >
> > > > Thank you, score revised to 7 (accept) owing to clarity of claims, as discussed.

---

### Official Review · Reviewer_Tdyv · 2021-07-18

**Rating:** 8
**Confidence:** 3

**Summary:**

This paper proposes a novel method to learn dense keypoints in a semi-supervised manner using unlabeled multiview images. They propose to leverage the epipolar constraint of dense correspondences given a known fundamental matrix. In order to make epipolar constraint learnable, they propose a probabilistic epipolar constraint, incorporating uncertainty in correspondences using soft correspondences. The proposed method is proven to improve dense keypoint detectors in multiple datasets over reasonable baseline methods.


**Limitations And Societal Impact:**

This paper thoroughly study the limitation of the proposed method.


**Main Review:**

The proposed method is reasonable, well written, and well studied. I cannot comment much about the originality of the work since this is out of my field of research, however the paper was easy to follow. Given that dense labeling of correspondence is not an easily affordable label, I have no doubt that self-supervised learning with known geometric constraints definitely helps.

Here are some minor comments that may potentially help me understand this work better:

It would be helpful to ablate the effect of L_R. Although L_R is supposed to be a regularization, there may be some knowledge distillation from the pretrained model on unlabeled dataset that may potentially help improve the keypoint accuracy by itself. L_L + L_R would make the effectiveness of L_M obvious, especially since L_L + L_M does not outperform L_L (although I believe is reasonable).

How does the choice of $\sigma$ in soft correspondence affect the performance of the model? Which value did you choose for $\sigma$? How did you find the optimal $\sigma$? Some ablation study on the effect of $\sigma$on the performance would be helpful.


**Time Spent Reviewing:**

3hrs

---

> ### Author Response · Authors · 2021-08-10
> **We address reviewer's concern on effectiveness of multi-view consistency loss and influence of sigma on model performance.**
>
> We are grateful for you providing insightful comments and potential improvement of the manuscript.
>
>  - The regularization loss ($L_R$) uses the pretrained model on labeled single view dataset as teacher network. It alone does not add any value: it is identical to the supervised model in theory, i.e., $L_L = L_L+L_R$. To empirically prove this claim, we conduct an additional ablation study by adding $L_L+L_R$ as shown in below Table. It can be seen that $L_L + L_R$ performs similarly to $L_L$ while $L_L + L_M + L_R$ outperforms $L_L + L_R$ and $L_L$ on all metrics (by a margin up to 6.9% for keypoint accuracy metrics and 9.7% for reconstruction accuracy metrics). Similarly, $L_L + L_M + L_R + L_T$ consistently outperforms $L_L + L_R + L_T$ on all metrics (by a margin up to 5.2% for keypoint accuracy metrics and 4.7% for reconstruction accuracy metrics), which further shows the effects of the epipolar loss.
>
> |     |                                                                                                     |           |           |           |           |            |           |           |
> |:----|:----------------------------------------------------------------------------------------------------|:----------|:----------|:----------|:---------:|:-----------|:----------|:----------|
> |     | Method                                                                                              | AUC\_10   | AUC\_30   | mRCI      |   mGPS    | Epi. error | MPVPE     | mMVS      |
> |     | Supervised ($\\mathcal{L}\_{\\rm L}$)                                                               | 0.48      | 0.748     | 0.777     |   0.864   | 5.54       | 58.71     | 0.521     |
> |     | $\\mathcal{L}\_{\\rm L} + \\mathcal{L}\_{\\rm M}$                                                   | 0.12      | 0.448     | 0.487     |   0.719   | 1.46       | 176.9     | 0.285     |
> |     | $\\mathcal{L}\_{\\rm L} + \\mathcal{L}\_{\\rm R}$                                                   | 0.482     | 0.749     | 0.779     |   0.864   | 5.36       | 58.47     | 0.513     |
> |     | $\\mathcal{L}\_{\\rm L} + \\mathcal{L}\_{\\rm M} + \\mathcal{L}\_{\\rm R}$                          | 0.513     | 0.76      | 0.782     |   0.867   | 2.18       | 55.16     | 0.563     |
> |     | $\\mathcal{L}\_{\\rm L} + \\mathcal{L}\_{\\rm R} + \\mathcal{L}\_{\\rm T}$                          | 0.499     | 0.755     | 0.779     |   0.866   | 4.65       | 52.97     | 0.570     |
> |     | $\\mathcal{L}\_{\\rm L} + \\mathcal{L}\_{\\rm M} + \\mathcal{L}\_{\\rm R} + \\mathcal{L}\_{\\rm T}$ | **0.525** | **0.764** | **0.783** | **0.868** | **2.13**   | **51.17** | **0.597** |
>
>  - $\sigma$ controls the range of neighbors. We use $\sigma=1.318e^{-2}$ in the manuscript. As suggested, we conduct a new ablation study as varying $\sigma$ as shown in Table below:
>
> |     |                                                                                                                                  |         |         |       |       |            |       |       |
> |:----|:---------------------------------------------------------------------------------------------------------------------------------|:--------|:--------|:------|:-----:|:-----------|:------|:------|
> |     | Method                                                                                                                           | AUC\_10 | AUC\_30 | mRCI  | mGPS  | Epi. error | MPVPE | mMVS  |
> |     | $\\mathcal{L}\_{\\rm L} + \\mathcal{L}\_{\\rm M} + \\mathcal{L}\_{\\rm R} + \\mathcal{L}\_{\\rm T} (\\sigma\\rightarrow\\infty)$ | 0.483   | 0.749   | 0.777 | 0.864 | 5.34       | 58.97 | 0.519 |
> |     | $\\mathcal{L}\_{\\rm L} + \\mathcal{L}\_{\\rm M} + \\mathcal{L}\_{\\rm R} + \\mathcal{L}\_{\\rm T} (\\sigma=2.402e^{-2})$        | 0.509   | 0.759   | 0.783 | 0.867 | 2.12       | 52.64 | 0.601 |
> |     | $\\mathcal{L}\_{\\rm L} + \\mathcal{L}\_{\\rm M} + \\mathcal{L}\_{\\rm R} + \\mathcal{L}\_{\\rm T} (\\sigma=1.908e^{-2})$        | 0.525   | 0.764   | 0.784 | 0.868 | 2.13       | 51.14 | 0.600 |
> |     | $\\mathcal{L}\_{\\rm L} + \\mathcal{L}\_{\\rm M} + \\mathcal{L}\_{\\rm R} + \\mathcal{L}\_{\\rm T} (\\sigma=1.576e^{-2})$        | 0.522   | 0.763   | 0.783 | 0.868 | 2.11       | 50.46 | 0.603 |
> |     | $\\mathcal{L}\_{\\rm L} + \\mathcal{L}\_{\\rm M} + \\mathcal{L}\_{\\rm R} + \\mathcal{L}\_{\\rm T} (\\sigma=1.318e^{-2})$        | 0.525   | 0.764   | 0.783 | 0.868 | 2.13       | 51.17 | 0.597 |
> |     | $\\mathcal{L}\_{\\rm L} + \\mathcal{L}\_{\\rm M} + \\mathcal{L}\_{\\rm R} + \\mathcal{L}\_{\\rm T} (\\sigma=1.084e^{-2})$        | 0.527   | 0.765   | 0.785 | 0.869 | 2.03       | 50.39 | 0.595 |
> |     | $\\mathcal{L}\_{\\rm L} + \\mathcal{L}\_{\\rm M} + \\mathcal{L}\_{\\rm R} + \\mathcal{L}\_{\\rm T} (\\sigma=9.320e^{-3})$        | 0.527   | 0.765   | 0.784 | 0.868 | 2.05       | 50.97 | 0.593 |
> |     | $\\mathcal{L}\_{\\rm L} + \\mathcal{L}\_{\\rm M} + \\mathcal{L}\_{\\rm R} + \\mathcal{L}\_{\\rm T} (\\sigma=7.610e^{-3})$        | 0.532   | 0.766   | 0.784 | 0.868 | 2.11       | 50.21 | 0.589 |

---

> > ### Comment · Reviewer_Tdyv · 2021-09-01
> > **Update**
> >
> > My concerns are addressed and I believe this paper would be a good contribution to the conference.

---

### Official Review · Reviewer_Muxb · 2021-07-19

**Rating:** 7
**Confidence:** 4

**Summary:**

This paper studies how to learn single image dense keypoint prediction by leveraging multiple-view constraints. Compared to prior works in this space, the key contribution is the modeling of the epipolar geometry constraint across different views.

**Ethical Concerns:**

NA.

**Limitations And Societal Impact:**

The limitations are not discussed explicitly, but the experimental results are analyzed in depth, including the pros and cons.

**Main Review:**

This paper studies an important problem of learning keypoint detectors by using multi-view supervision. Compared to prior works, the key contribution is the learning of dense keypoints and using epipolar constraints.

The epipolar constraint provides some supervision for the dense keypoint prediction. However, it is not a direct supervision as we do not know which point on the epipolar line is the corresponding point. This paper proposes a solution, which is similar to an attention mechanism, by matching points in some feature space. The proposed solution is plausible.

The network training incorporates other losses such as photometric multi-view consistency and regualrizations using a pre-trained model.

The experimental results show that the proposed approaches outperform state-of-the-art approaches on two benchmark datasets.

There are a couple of issues that prevent me from giving higher scores:
1. It is not entirely clear how to address the partial similarity across the multiple views. The paper does mention that this is an issue. However, it is unclear how the definition of $P(u,u')$ and equations (2) and (3) address this.
2. The epipolar constraints state that two lines of the two images are in correspondence, and correspondences are expected to be continuous. The current formulation in the paper only use such signals partially.
3. It is unclear how to the canonical model is defined. Figure 2 shows that it may be a 2D surface parameterization of a 3D mesh. If so, why the phi map is not invertible?
4. The current setup requires calibrated images with known intrinsics and extrinsics. What about if the estimations of those have errors and inaccurate.


**Time Spent Reviewing:**

1 hour

---

> ### Author Response · Authors · 2021-08-10
> **We address reviewer's question on partial similarity, continuity of correspondences, UV mapping and calibration.**
>
> We are grateful for you providing insightful comments and potential improvement of the manuscript.
>
>  1. We suppose the term “partial similarity” here means “visibility” in the manuscript (L142-143 & equation (4)). We will add the following clarification:
> “The visibility is computed on-the-fly using predicted dense keypoints $\phi$. It is visible if there exists a corresponding $\mathbf{u}’ = \phi(\mathbf{y}; \mathcal{I}’)$ from view $\mathcal{I}’$ to the predicted $\mathbf{u} = \phi(\mathbf{x}; \mathcal{I})$ from image I, i.e., $\exists \mathbf{u}' \in \Omega, \|\mathbf{u}’ - \mathbf{u}\| < \epsilon$ where $\epsilon$ is correspondence tolerance defined in the canonical UV space." We use $\epsilon=0.03$ in the paper. We conduct an additional study on two different tolderances 0.02 and 0.05 and report results in the following table. It can be seen that the performance for those 3 different thresholds are similar, but models trained with threshold 0.03 outperform the other two on AUC_10 metric by a margin of 1.3%.
>
> |     |                                                                                                                        |           |           |           |           |            |           |           |
> |:----|:-----------------------------------------------------------------------------------------------------------------------|:----------|:----------|:----------|:---------:|:-----------|:----------|:----------|
> |     | Method                                                                                                                 | AUC\_10   | AUC\_30   | mRCI      |   mGPS    | Epi. error | MPVPE     | mMVS      |
> |     | Supervised ($\\mathcal{L}\_{\\rm L}$)                                                                                  | 0.48      | 0.748     | 0.777     |   0.864   | 5.54       | 58.71     | 0.521     |
> |     | $\\mathcal{L}\_{\\rm L} + \\mathcal{L}\_{\\rm M}$                                                                      | 0.12      | 0.448     | 0.487     |   0.719   | 1.46       | 176.9     | 0.285     |
> |     | $\\mathcal{L}\_{\\rm L} + \\mathcal{L}\_{\\rm M} + \\mathcal{L}\_{\\rm R}$                                             | 0.513     | 0.76      | 0.782     |   0.867   | 2.18       | 55.16     | 0.563     |
> |     | $\\mathcal{L}\_{\\rm L} + \\mathcal{L}\_{\\rm M} + \\mathcal{L}\_{\\rm R} + \\mathcal{L}\_{\\rm T}$                    | **0.525** | **0.764** | **0.783** | **0.868** | **2.13**   | **51.17** | **0.597** |
> |     | $\\mathcal{L}\_{\\rm L} + \\mathcal{L}\_{\\rm M} + \\mathcal{L}\_{\\rm R} + \\mathcal{L}\_{\\rm T} (\\epsilon = 0.02)$ | 0.518     | 0.762     | 0.783     |   0.868   | 2.14       | 51.38     | 0.593     |
> |     | $\\mathcal{L}\_{\\rm L} + \\mathcal{L}\_{\\rm M} + \\mathcal{L}\_{\\rm R} + \\mathcal{L}\_{\\rm T} (\\epsilon = 0.05)$ | 0.519     | 0.762     | 0.783     |   0.868   | 2.12       | 51.87     | 0.603     |
>
> 2. We do not use spatial continuity for making correspondence. It is a great suggestion that motivates our future work. We will include this in our Discussion section.
>
> 3. We use the canonical model defined by dense human keypoint detection methods (i.e., DensePose). It maps a pixel to a UV texture coordinate that is pre-defined by a 3D mesh model (e.g., SMPL). Mapping from pixels to UV space is injective where the inverse does not exist (e.g., there exists UV coordinates that do not map to pixels because of occlusion).
>
> 4. Our method requires camera calibration similar to multiview supervision approaches, e.g. Bootstrapping[22] and MONET[50]. Error in calibration cannot be addressed by our method (Section B of supplementary material).

---

> > ### Comment · Reviewer_Muxb · 2021-08-25
> > **Concerns are addressed**
> >
> > I support accepting this paper.

---

### Decision · Program_Chairs · 2021-09-27

**Decision:**

Accept (Spotlight)

**Comment:**

Reviewers agreed that this is a solid paper that deserves acceptance. Authors are highly encouraged to address the key comments reported by reviewers as well as to implement all the improvements (as indicated by authors in the rebuttal) in the final camera-ready version.